# LVD-2M: A Long-take Video Dataset with Temporally Dense Captions

**Tianwei Xiong**[1]* **Yuqing Wang**[1]* **Daquan Zhou**[2]† **Zhijie Lin**[2] **Jiashi Feng**[2] **Xihui Liu**[1]✉

[1]The University of Hong Kong    [2]ByteDance
https://silentview.github.io/LVD-2M/

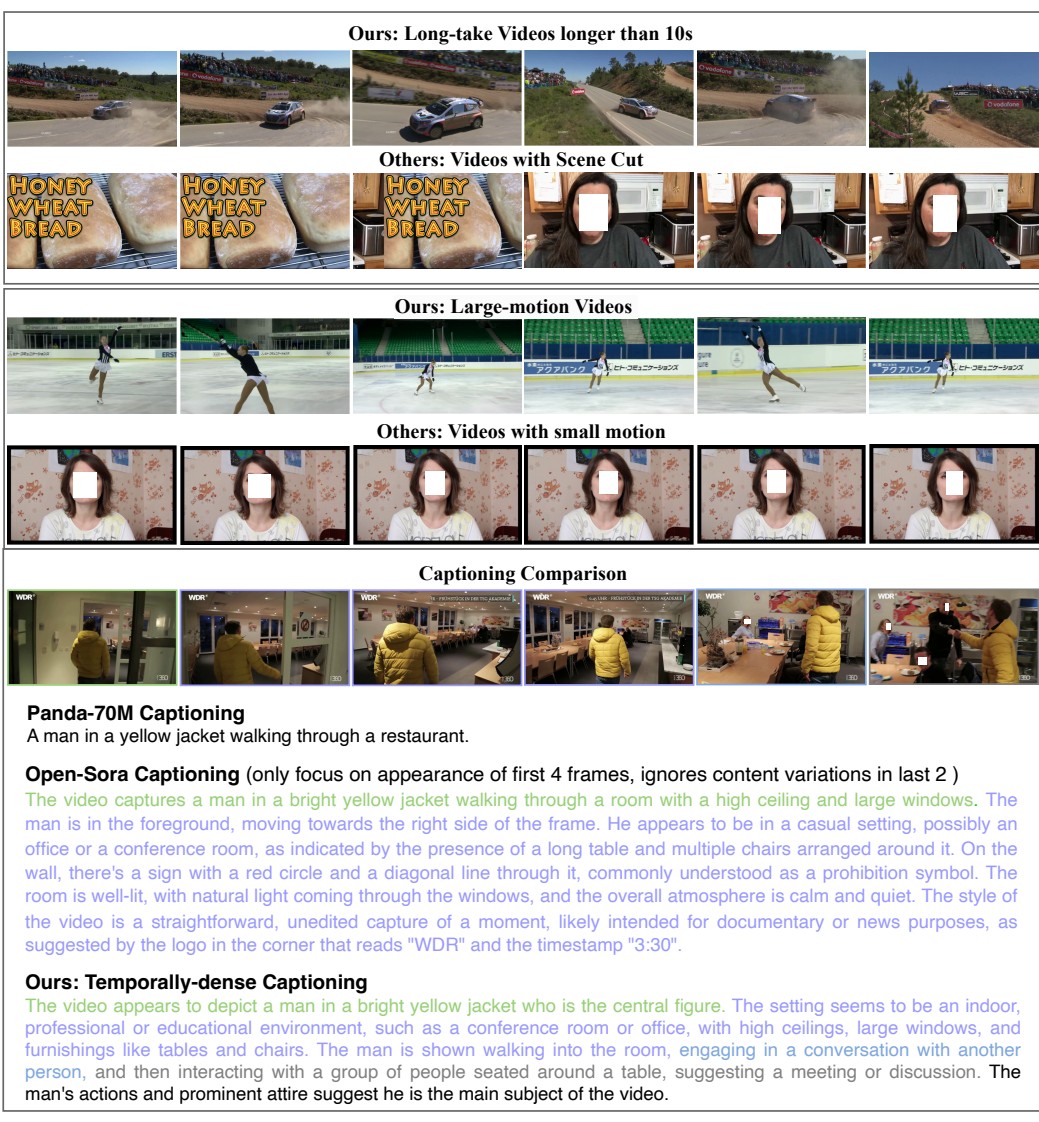

Figure 1: Comparison of our proposed LVD-2M dataset against previous datasets. Our dataset contains long-take videos with significant motion and temporally-dense captions (different colors represent captions for different frames), contrasting with short videos and sparse annotations in previous datasets like Panda-70M [1], HD-VG [2], and WebVid [3] (shown as "Others").

*Equal contributions.  † Project lead.  ✉Corresponding author.

38th Conference on Neural Information Processing Systems (NeurIPS 2024) Track on Datasets and Benchmarks.

## Abstract

The efficacy of video generation models heavily depends on the quality of their training datasets. Most previous video generation models are trained on short video clips, while recently there has been increasing interest in training long video generation models directly on longer videos. However, the lack of such high-quality long videos impedes the advancement of long video generation. To promote research in long video generation, we desire a new dataset with four key features essential for training long video generation models: (1) long videos covering at least 10 seconds, (2) long-take videos without cuts, (3) large motion and diverse contents, and (4) temporally dense captions. To achieve this, we introduce a new pipeline for selecting high-quality long-take videos and generating temporally dense captions. Specifically, we define a set of metrics to quantitatively assess video quality including scene cuts, dynamic degrees, and semantic-level quality, enabling us to filter high-quality long-take videos from a large amount of source videos. Subsequently, we develop a hierarchical video captioning pipeline to annotate long videos with temporally-dense captions. With this pipeline, we curate the first long-take video dataset, LVD-2M, comprising 2 million long-take videos, each covering more than 10 seconds and annotated with temporally dense captions. We further validate the effectiveness of LVD-2M by fine-tuning video generation models to generate long videos with dynamic motions. We believe our work will significantly contribute to future research in long video generation.

## 1  Introduction

Generating long-take videos with temporal consistency, rich contents and large motion dynamics is essential for various applications such as AI-assisted film production. Although video generation models [4–8] have achieved impressive results in generating short video clips of few seconds, it remains challenging to simulate temporal-consistent and dynamic contents over long durations. Some works [9–11] attempt to extend video generation models trained on short video clips to long video generation by iteratively generating next frames conditioned on previously generated frames. However, those methods suffer from temporal inconsistency and limited motion patterns. Inspired by Sora [12], there has been increasing interest in scaling up video generation models for longer videos [13, 14]. Being trained directly on long-duration videos, these models provide a promising path toward modeling long-range temporal consistency and large motion dynamics in long videos. However, an obstacle on this path is the lack of high-quality long videos with rich text annotations.

Previous datasets of large-scale video-text pairs [3, 2, 15] have made significant contributions to video generation, but most of them encounter limitations for training long video generators. Video datasets crawled from the Internet [2, 3, 15] usually contain static videos or scene cuts, which are harmful to the training of video generation models. Moreover, previous text-to-video generation datasets are annotated with only short video captions, failing to capture the rich and dynamic semantics in long videos. Despite several recent efforts [12, 14] in generating long captions for videos, they mostly focus on generating spatially-dense captions and neglect the rich temporal dynamics in videos.

It has been validated in previous works [16] that fine-tuning pre-trained generative models on high-quality datasets could significantly improve the quality of generated images and videos. Despite previous efforts in building large-scale video datasets, high-quality long video datasets with dense annotations are rarely available and expensive. Inspired by this, we desire a dataset specifically designed for long video training with the following properties: (1) long videos covering at least 10 seconds, (2) long-take videos without cuts, (3) large motion and diverse content, and (4) annotated with temporally-dense captions.

To this end, we create an automatic pipeline for video filtering and long video recaptioning. We devise a video filtering process leveraging both low-level filtering tools including scene cut detection and optical flow [17] estimation, and semantic-level filtering tools like video LLMs [18]. The video filtering process selects high-quality long-take videos spanning over 10 seconds without scene cuts and containing large motion dynamics. Moreover, we design a hierarchical captioning approach to generate temporally-dense captions for long videos. Specifically, we split long videos into 30-second clips. For each clip, we uniformly sample 6 frames and arrange them in a grid layout. The single

composite image, named "image grid" [19], is fed into LLaVA-v1.6-34B [20] for temporally-aware video clip captioning. Then, we apply a Large Language Model, Claude3-Haiku [21], to refine the captions and integrate captions from different clips into a complete caption describing the whole video. Compared to captions of previous video datasets, our hierarchical captioning approach provides temporally-dense captions describing the transitions of actions and scenes over the whole duration.

Following our pipeline, we generate 2 million high-quality video-caption pairs from 220 million videos in 4 open-sourced large-scale datasets: Panda-70M [1], HD-VG-130M [2], InternVid [22], and WebVid-10M [3]. Human evaluations demonstrate that our dataset is preferred by human raters in terms of dynamic degree, long-take videos without scene cuts, and quality of captions. We further validate the effectiveness of our LVD-2M by fine-tuning pre-trained video generation models on LVD-2M. We experiment on both diffusion-based video generation models and language model-based video generation models. We find that models fine-tuned on this dataset perform better at generating long videos with large motion dynamics. Moreover, the model learns to generate long-take videos with significant camera movement accompanied by smooth scene transitions.

In summary, our contributions are three-fold. **1)** We devise an automatic data curation pipeline, including low-level and semantic-level filtering strategies to select high-quality long-take videos with large motions, and a hierarchical captioning approach to annotate long videos with temporally-dense captions. **2)** To address the lack of high-quality data for long video generation, we leverage our proposed data curation pipeline to construct LVD-2M, a dataset of high-quality long-take videos spanning over 10 seconds, with temporally-dense captions. **3)** We validate the effectiveness of LVD-2M by both human evaluation and fine-tuning experiments on both diffusion-based and LM-based video generation models using LVD-2M.

## 2   Related Work

**Video-Language Datasets.**   To effectively train video generative models, a high-quality video-language dataset is crucial. Early datasets, such as MSR-VTT [23] and ActivityNet [24], were created through manual annotation, which limited their scale. Subsequent works aimed to increase dataset scale by utilizing automatic speech recognition (ASR) to extract text descriptions from videos. Notable examples include HowTo100M [25], YT-Temporal [26], and HD-VILA [15]. Although this approach significantly increased the amount of data, the ASR-generated text descriptions often fail to accurately represent the main video content. Another approach is to directly use readily available titles or descriptions of online videos as captions. WebVid [3] followed this approach and collected 10 million video-text pairs, primarily from stock footage providers. A common limitation of existing datasets is that the vast majority of samples are short video clips, lacking coverage of long videos, especially dense descriptions of long-range dynamic content changes. For dataset targeting longer vidoes, StoryBench [27] has provided a few thousand annotated long videos, but its limited data scale restricts its usage to evaluation rather than model training. A concurrent work ShareGPT4Video [28] curated a dataset with long videos and detailed captions, but its data pipeline is less focused on video data filtering and processing. To truly drive advances in long video generation models, constructing a large-scale dataset of high-quality long-take videos with dense captions is crucial.

**Video Generation.** Most existing video generation methods primarily focus on generating short video clips, with diffusion models  [5–7] being the prevalent approach. There are also a few works based on language models (LM-based) [8] for video generation. Some works attempt to extend to long video generation by training models on short video data and then employing techniques such as sliding window generation [11, 10, 29, 9]. However, these methods often suffer from quality degradation, lack of temporal consistency, and difficulty in generating high-quality long-range dynamic video content. We identify that a lack of high-quality long video datasets hinders existing text-to-video generative models from effectively modeling and generating long videos with rich dynamics.

**Video Understanding.** Vision-language [19, 18, 30, 31] models demonstrated strong performance in video understanding. Recently, IG-VLM [19] pointed out that an VLM [20] comprehensively pretrained on images can be highly capable of video understanding. This is achieved by concatenating multiple frames from a video into a single image in grid view, which will be the input for VLMs. In this work, we propose a way to filter undesired videos utilizing a Video-LLM [18] which can largely enhance the overall quality of the dataset.

**Video Captioning.** The usage of VLMs for video understanding has been primarily focused on VQA tasks [32, 30]. But it is less explored specifically for video captioning. Previously, HD-VG [2] utilizes BLIP-2 [33] to caption a single key frame from a video clip, Panda-70M [1] trains a light-weighted captioning model for captioning, and InternVid [22] combines BLIP-2 captions for multiple frames into a single overall caption with a language model. The resulted captions from these previous caption pipelines are mostly a single sentence. In this work, we target on the generation of detailed and temporally dense captions that better capture the content of the videos, utilizing a strong VLM [20].

## 3 Dataset

We devise a data curation pipeline to filter large-motion long-take videos from large-scale video datasets and to annotate them with temporally-dense captions. We demonstrate the data curation pipeline and data statistics of LVD-2M in this section.

### 3.1 Long-take Video Collection and Filtering

**Collecting videos from source datasets.** We collect videos from four sources: (1) HD-VG [2] which contains 130 million video clips collected from YouTube. (2) InternVid [22] which contains 38 million video clips from YouTube. (3) Panda70M [1] which contains 70 million videos from YouTube. (4) WebVid [3] which contains 10 million videos from stock footage providers. However, not all of those videos are suitable for long video generation. For example, only 15% of video clips from InternVid [22] are longer than 10s, while around 52.5% of these long videos contain shot changes (Tab. 2). While videos from stock footage providers [3] seldom contain scene cut, nearly half of these videos are not dynamic (Fig. 5). Those low-quality videos will hinder the training of long video generation models. Thus, we devise several filtering criteria to select high-quality, large-motion, and long-take videos from 220 million videos in the four datasets. The whole filtering process is shown in Fig. 2.

**Selecting long-take videos with scene cut detection.** Most current video generation models are trained on short video clips, and videos crawled from the Internet contain many scene cuts, which may impede the long video generation models from learning long-range temporal consistency and continuous motion across frames. We aim to select videos of consistent scenes captured over 10 seconds. It is worth mentioning that smooth transition of scenes (*e.g.*, the background of a street continuously changes as a person walks down the street) is allowed, and we only target filtering out scene cuts or slow shot changes with fade-in and fade-out effects caused by post-editing of videos. Previous attempts [1, 2, 22] leverage PySceneDetect [34] to detect sudden shot changes and semantic consistency [1] between early and late frames to detect large scene changes. However, there is still a portion of videos with fade-in / fade-out shot changes in the filtered datasets. We optimize the settings of PySceneDetect to better detect both sudden scene cuts and slow shot changes with fade-in / fade-out effects. Specifically, we find that the default setting $\mathrm{AdaptiveDetector}$ with a rolling average threshold leads to difficulties in detecting slow shot changes with fade-in and fade-out effects. To filter out both sudden and slow scene cuts, we use $\mathrm{ContentDetector}$ with $\mathrm{cutscene\_threshold}$ of 50 and $\mathrm{min\_scene\_len}$ of 0 frames on video frames sampled at a low fps of 0.5. By applying PySceneDetect on the whole video, videos with any significant changes within a 2-second interval are filtered out, including fade-in and fade-out effects which are commonly within 2 seconds.

**Selecting large-motion videos with optical flow.** We use optical flow as a clue to filter out static videos with little motion dynamics. Specifically, we calculate the optical flow with RAFT [17] between each pair of neighboring frames sampled at 2 fps and discard any videos with an average optical flow magnitude below a threshold of 20. This step helps remove videos with minimal motion, such as static scenes or individuals speaking to the camera against a still background.

**Removing low-quality videos with MLLMs.** We further conduct semantic-level filtering with MLLMs to remove low-quality videos that cannot be detected by previous filtering strategies. We leverage the PLLaVA-7B [18], which extends LLaVA from images to videos, for semantic-level filtering. For each video, we uniformly sample 8 frames from each video and prompt PLLaVA to distinguish low-quality videos. Specifically, we filter out videos that lack diversity, lack content variations, or with low perceptual qualities. The optical-flow-based criteria in the previous step can

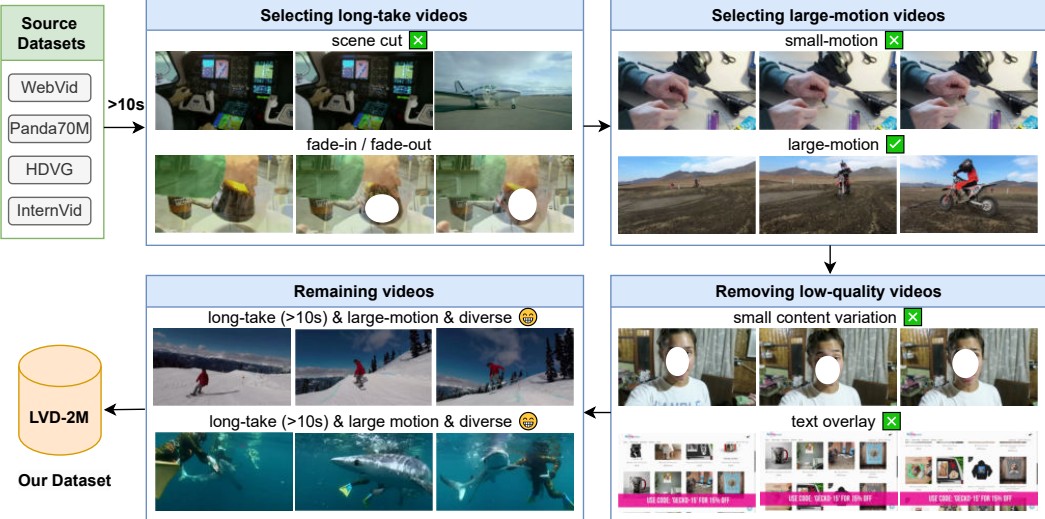

Figure 2: Video filtering process. Our video filtering process employs multiple criteria to select high-quality, dynamic, and long-take videos from four source datasets.

filter out most near-static videos. However, some shaky videos captured by hand-holding cameras achieve high optical flow scores despite their lack of meaningful motion. Thus, we leverage PLLaVA to distinguish those low-quality videos. We further filter out videos with extensive text overlays because those videos add extra burdens for model training.

## 3.2 Hierarchical Long Video Captioning for Temporally Dense Captions

We propose a hierarchical captioning approach to annotate temporally-dense captions for long videos. As shown in Fig. 3, we first split videos longer than 30 seconds into video clips of 30 seconds. Then we annotate the clip-level video captions for each video clip. Finally we use an LLM to refine the captions and merge captions from all clips into a temporally-dense caption for the whole video. In this subsection, we first demonstrate how to caption video clips shorter than 30 seconds, and then demonstrate how to use LLM to refine and merge captions.

**Captioning a video clip as an image grid.** A recent work [19] has demonstrated that Vision Language Models (VLMs) pretrained only on images have strong zero-shot performance in video understanding. We generate captions for video clips shorter than 30 seconds inspired by this approach. Specifically, we uniformly sample 6 frames from the video clip and arrange these frames into a single composite image with a grid layout. We then input the image grid to LLaVA-v1.6-34B [20] to generate the video clip captions. With this approach, we can obtain detailed captions describing the backgrounds, main characters, major actions, and camera perspectives in the video clips.

**Refining and merging captions with LLMs.** We identify that solely applying VLMs may not be sufficient for generating high-quality captions. LLaVA-v1.6-34B is prone to generating extra interpretations or assumptions about videos, leading to redundancy in the generated captions. So we leverage an LLM, Claude3-Haiku [21], to further refine the generated captions. In particular, we prompt Claude3-Haiku to rewrite the given raw captions so that the new captions are concise, objective, and convey a clear storyline for the video. Furthermore, for videos longer than 30 seconds, we prompt Claude3-Haiku to compose the multiple captions into a single, coherent caption describing the content and dynamics of the whole video.

## 3.3 Dataset Statistics

We present the comparison between our LVD-2M and previous video datasets in Tab. 1. LVD-2M is a high-quality dataset with videos longer than 10 seconds. Compared to previous video datasets, videos in LVD-2M are with large motion and rich captions. We further present the statistics of the category distribution, duration, and word count of our dataset in Fig. 4. To understand the

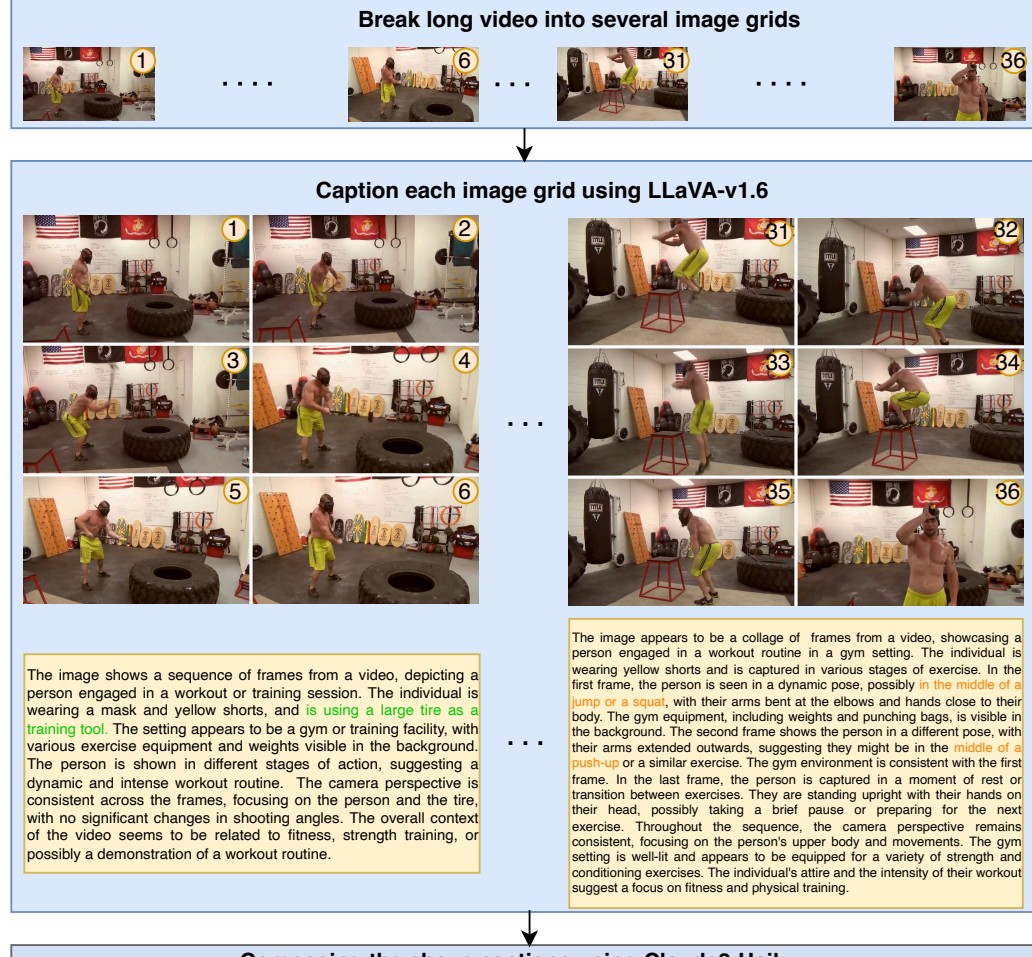

Figure 3: Hierarchical video captioning process. First, we split the long video into 30-second clips and compose them into image grids. Then, we use the LLaVA-1.6 model [20] to generate captions for each image grid. Finally, we use the Claude3-Haiku model [21] to refine and merge these captions into the final complete caption for the whole video.

distribution of collected video categories, we utilize the BART model [36] to classify the video captions into 8 categories based on the main objects and content. As shown in Fig. 4, our dataset covers diverse categories commonly found in the real world, such as scenery, people, food, sports, animals, transportation, gaming, and others.

# 4    Experiments

In Sec. 4.1, we conduct human evaluation analysis to demonstrate that our filtered video dataset, LVD-2M, contains fewer scene cuts, larger motion dynamics, and higher-quality captions, compared with previous datasets. In Sec. 4.2, we further validate the effectiveness of our LVD-2M by fine-tuning pre-trained video generation models on LVD-2M. We conduct fine-tuning experiments on both diffusion-based video generation models and language model-based video generation models, and find that fine-tuning video generation models on our dataset boosts the video generation models' abilities in generating long-take videos with large motion dynamics. In Sec. 4.3, we present the

Table 1: Comparison of LVD-2M and other video datasets.

| Dataset | Text | Avg/min video len | | Avg text len | Avg optical flow score (>10s) |
|---|---|---|---|---|---|
| HowTo100M [25] | ASR | 3.6s | - | 4.0 words | - |
| ACAV [35] | ASR | 10.0s | - | - | - |
| YT-Temporal-180M [26] | ASR | - | - | - | - |
| HD-VILA-100M [15] | ASR | 13.4s | - | 32.5 words | - |
| Panda-70M [1] | Automatic caption | 8.5s | <1s | 13.2 words | 14.7 |
| HD-VG-130M [2] | Automatic caption | 5.8 s | <1s | 9.8 words | 21.5 |
| WebVid-10M [3] | Scrapped Footage Caption | 18.0s | <1s | 14.1 words | 12.1 |
| InternVid-38M [22] | Automatic caption | 17.2s | <1s | 17.6 words | 11.6 |
| **LVD-2M (Ours)** | Temporally dense caption | 20.2s | 10s | 88.7 words | 47.8 |

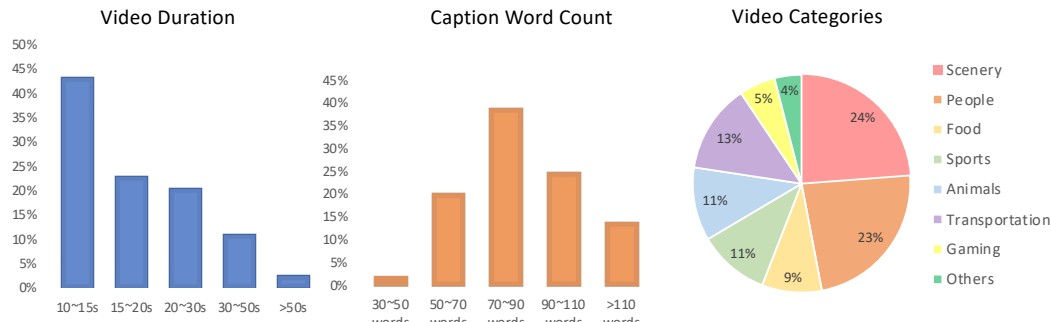

Figure 4: Statistics of LVD-2M. LVD-2M consists of long video clips with detailed dense captions, and diverse categories.

effectiveness of LVD-2M to extend the generation frame length of a diffusion-based T2V model, with comprehensive quantitative and qualitative validations.

## 4.1 Human Evaluation of Dataset Quality

To validate the quality of LVD-2M and the effectiveness of our data curation pipeline, we conduct human evaluations to examine the long-take consistency, dynamic degrees, and caption qualities. For human evaluations, we compare our LVD-2M with previous video datasets: Panda-70M [1], HD-VG-130M [2], InternVid [22], and WebVid-10M [3].

**Long-take consistency in videos.** We examine that the filtered videos are mostly long-take videos without cuts. We randomly sample 40 videos from each dataset, each one being 10~30s long. We do not compare with WebVid [3] because its videos are from stock footage providers and barely have scene cuts.

Table 2: Long-take video clip ratio, based on human raters, comparing LVD-2M with other video datasets.

| InternVid | Panda-70M | HD-VG | LVD-2M |
|---|---|---|---|
| 47.5% | 50.0% | 55.0% | 77.5% |

For fair comparison, we also exclude videos collected from WebVid in samples from LVD-2M. The sampled videos are mixed and randomly shown to human raters. We request human raters to check for any type of scene cut that can lead to inconsistency. As shown in Tab. 2, with our video filtering strategy, LVD-2M reaches the highest long-take video ratio. We examine the cases in our dataset deemed by human raters as non-long-take videos, and identify the major failure cases are slight jump cuts. While humans can easily recognize a slight jump cut in a video, it is challenging for scene cut detection algorithms and MLLM-based semantic-level filtering models to identify such slight changes in the videos.

**Dynamic degree of videos.** We randomly sample 40 videos for each dataset, each one being 10~30s long. We request human raters to rate the dynamic degree of the given videos from 1 to 3, where 1 means being not dynamic and 3 for being very dynamic. As shown in Fig. 5, for previous datasets, a large portion of videos are considered as not dynamic. After filtering at low-level with optical flow scores and at high-level with MLLMs, our LVD-2M successfully get rid of most static videos and the achieve a larger portion of very dynamic videos.

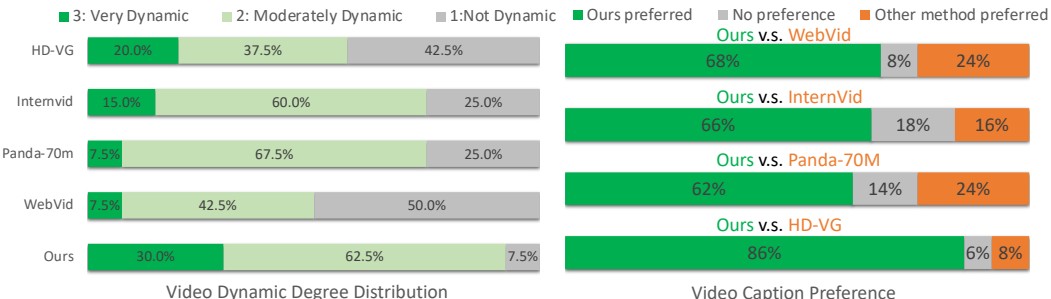

Figure 5: The distribution of human-rated dynamic degree score and human preference for caption quality, comparing LVD-2M with other video datasets.

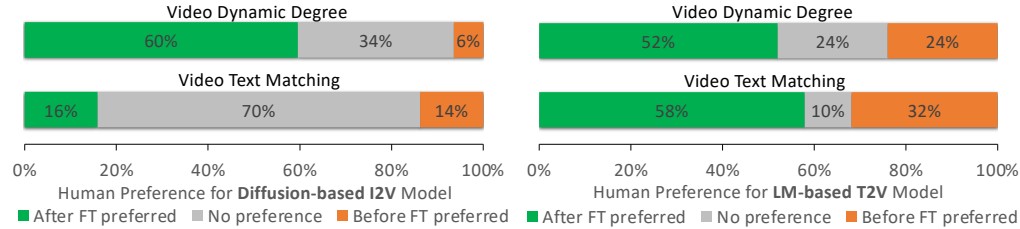

Figure 6: Human evaluation of generated videos by baseline v.s. fine-tuned models. We finetune both a diffusion-based I2V model and a LM-based T2V model on LVD-2M. Compared to the pretrained model, the finetuned models can generate more dynamic videos.

**Quality of video captions.** To compare the quality of our new captions to the original captions from datasets, we randomly sample 50 videos from each dataset. For each task, human raters are presented with a video clip and two captions, one from our captioning strategy, and another from the original dataset that the video is filtered from. We ask the human raters to compare the quality of the two captions. As shown in Fig. 5, our temporally-dense video captions are much more preferred by human raters. Among our baselines, Panda-70M captioning model [1] shows the best performance. As shown in Tab. 1, our captions are much longer and contain more details than previous datasets.

### 4.2 Fine-tuning Video Generation models with LVD-2M

To further validate the effectiveness of our LVD-2M in fine-tuning video generation models for generating long videos with large motion dynamics, we conduct fine-tuning experiments on a diffusion-based image-to-video (I2V) generation model and a language model-based text-to-video (T2V) generation model for long video generation. In this experiment, we don't extend the generation frame length of the pretrained models and compare the finetuned models with the pretrained ones. We further compare LVD-2M to WebVid-10M on extending the generation frame length for a diffusion-based T2V models in Sec. 4.3, and for a diffusion-based I2V model in the Appendix.

**Fine-tuning an LM-based T2V model.** We finetune a 7B LM-based video generation model from Loong [37]. The model utilizes a discrete video tokenizer similar to MAGVIT-v2 [38] to convert videos into tokens, and then models the video tokens with decoder-only autoregressive transformer. The model is pretrained on 15 million video-text pairs for 500K iterations with a batch size of 256. We further fine-tune it for 10k iteration with a batch size of 256 on 65-frame clips from LVD-2M.

**Fine-tuning a diffusion-based I2V model.** We finetune an I2V model, which was pretrained to generate 17-frame videos on 19 million video-text pairs for 18k iterations with a batch size of 288, following the similar image conditioning settings as proposed in EMU [16]. The model follows a similar architecture as MagicVideo [39] with 1.8B parameters.

**User study.** To validate the performance improvement after fine-tuning on LVD-2M, we conduct a user study comparing the pretrained models and finetuned models. For each base model (diffusion-based I2V and LM-based T2V), we use 50 text prompts to generate videos with the pretrained and fine-tuned models, respectively. Human raters are presented with 2 videos generated by the

Table 3: VBench evaluation for the two finetuned diffusion-based T2V models on LVD-2M and WebVid-10M [3] separately. Metrics exhibiting an absolute difference greater than 8% between the two models are underlined for emphasis.

| Finetuning Dataset | Subject Consistency | Background Consistency | Temporal Flickering | Motion Smoothness | Dynamic Degree | Aesthetic Quality | Imaging Quality | Object Class |
|---|---|---|---|---|---|---|---|---|
| WebVid-10M | 95.81% | **98.02%** | **98.00%** | 97.87% | 20.00% | **58.02%** | **72.63%** | 76.95% |
| LVD-2M | **96.12%** | 96.92% | 97.44% | **98.43%** | **28.06%** | 57.56% | 70.72% | **86.93%** |

| Finetuning Dataset | Multiple Objects | Human Action | Color | Spatial Relationship | Scene | Appearance Style | Temporal Style | Overall Consistency |
|---|---|---|---|---|---|---|---|---|
| WebVid-10M | **26.02%** | 61.40% | 75.51% | 51.06% | 29.19% | 20.12% | 19.34% | **21.43%** |
| LVD-2M | 22.76% | **76.20%** | **79.32%** | **51.40%** | **32.95%** | **20.60%** | **20.25%** | 21.29% |

pretrained and finetuned models respectively, conditioned on the same text. They are asked to choose the preferred video based on either video-text alignment or dynamic degree of generated videos. We collect 200 valid responses from human raters. As show in Fig. 6, we observe fine-tuning the LM-based T2V model on LVD-2M boosts the model's performance in terms of generating more dynamic videos and better alignment between generated videos and text prompts. On the other hand, after fine-tuning the diffusion-based I2V model on LVD-2M, the generated videos are significantly more preferred by users in terms of dynamic degree, with the win rate of 60% v.s. 6%. Although for diffusion-based I2V model, the improvement for video-text matching is relatively small, we identify that this may originate from the use of frozen clip text encoder for encoding long captions (88.65 words on average), since the maximum encoding length for clip text encoder is 77 tokens, and CLIP text encoder is not good at understanding long text prompts.

### 4.3 Extending a Diffusion-based T2V Model for Longer Range on LVD-2M

In this section, we present the effectiveness of LVD-2M for finetuning text-to-video (T2V) diffusion models to generate longer and more dynamic videos. For comparison, we choose the widely adopted WebVid-10M [3] as the baseline. In the experiment, we extend a T2V diffusion model from pretrained 32-frame generation length to 65-frame length, using LVD-2M and WebVid-10M seperately. Quantitative results on VBench [40] and qualitative comparisons can both validate the superiority of LVD-2M.

**Setup.** We finetune a base T2V diffusion model with 1.75B parameters, which has a similar structure as MagicVideo [39]. The base model was pretrained to generate 32-frame videos and finetuned at 65-frame length in this experiment. The finetuning settings for LVD-2M and WebVid-10M are the same, which is 64 batch size, 4 gradient accumulation iterations and for 30k iterations, roughly going over 2M video clips once at the finetuning stage. For quantitative evaluation, we follow the standard evaluation protocol of VBench [40].

**Results and analysis.** As shown in Tab. 3, compared to WebVid-10M, finetuning on LVD-2M will lead to better performance in 10 out of 16 metrics of VBench, especially surpassing WebVid-10M by a large margin in **dynamic degree**, **object class** and **human action**. These obvious performance improvements against the baseline can be attributed to the diverse and the highly dynamic video data of LVD-2M. Notably, the evaluation prompts from VBench have a small average length (7.6 words), which is much closer to the average caption length of WebVid-10M (14.1 words) than LVD-2M (88.7 words). Despite the caption length gap between training and evaluation, the model finetuned on LVD-2M still presents superior overall performance. We further demonstrate the qualitative comparisons in Fig. 7. Due to limited computational resources, we didn't validate LVD-2M on stronger T2V models, and the text encoding of the chosen T2V model is still based on CLIP [41], which struggles to properly encode long captions. We expect even more obvious performance enhancement when finetuning on LVD-2M using more advanced T2V models with more powerful text encoders [42].

## 5 Conclusion

High-quality long video datasets are essential for training long video generation models. In this work, we devise an automatic data curation pipeline to filter high-quality long-take videos from existing large-scale video datasets and to annotate temporally-dense captions for the filtered videos. Based on this pipeline, we construct LVD-2M, the first long-take video dataset of 2 million videos with large motion, diverse content, and temporally dense captions. We validate the quality of the dataset through human evaluation and verify its effectiveness by fine-tuning video generation models to generate long videos with large motions.

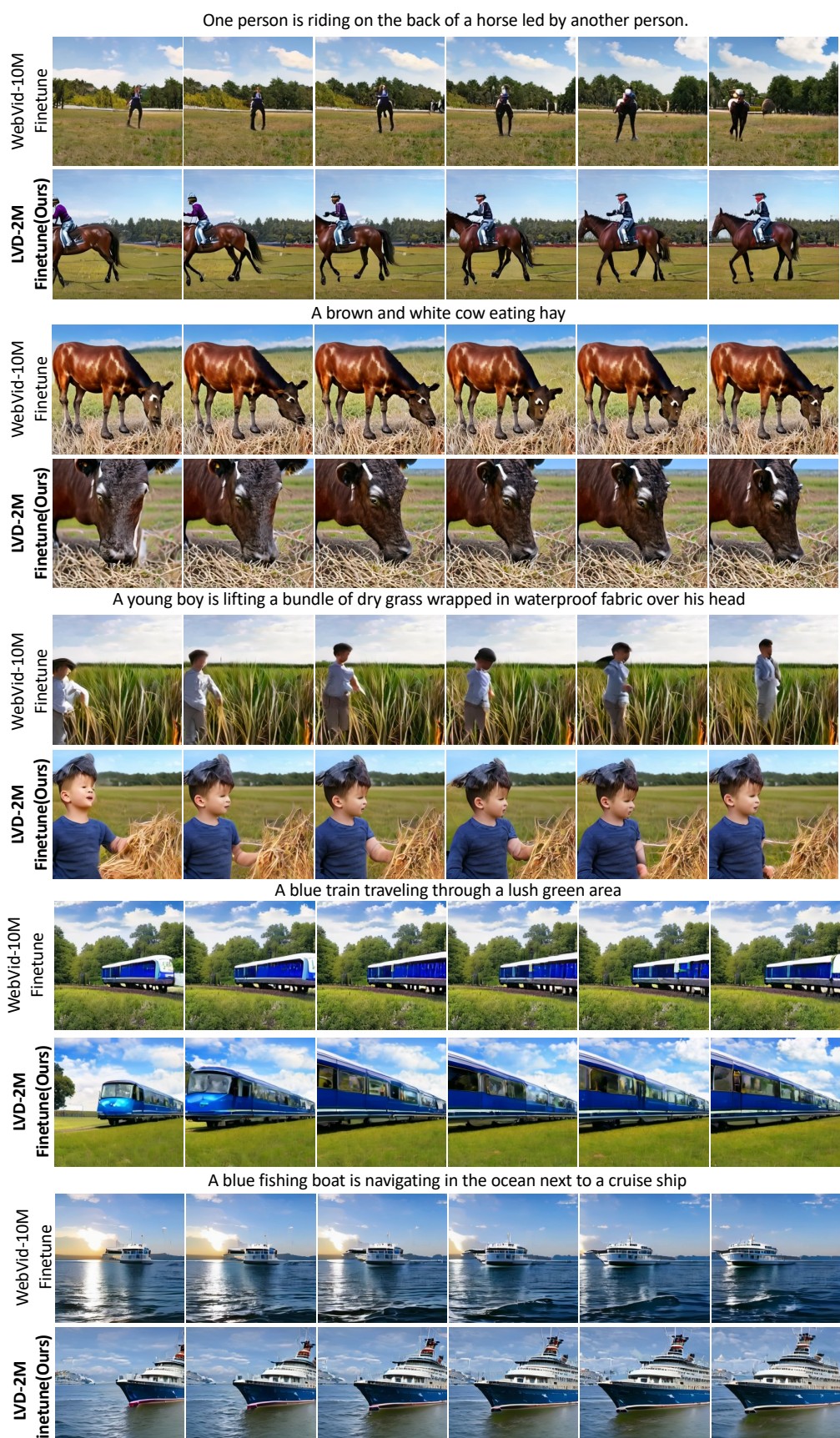

Figure 7: After finetuning a T2V diffusion model on LVD-2M, the videos are more dynamic, and the actions and objects in the videos are more reasonable, in contrast to finetuning on WebVid-10M.

## Acknowledgements

This work is supported in part by HKU Startup Fund, HKU Seed Fund for Basic Research, HKU Seed Fund for Translational and Applied Research, HKU IDS research Seed Fund, and HKU Fintech Academy R&D Funding.

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

## A  Limitations and Social Impacts

A limitation of our work is that the size of 2 million video-text pairs is not as large as other video datasets. However, those 2 million videos are high-quality videos filtered from 220 million videos tailored for long video generation. We will keep maintaining the dataset and expand the scale of the dataset in future versions. Our proposed dataset can be used to fine-tune video generators for long video generation. The resulting video generation models can be deployed to assist various applications such as film production. However, the community should be aware of the potential negative social impact that video generators may be used for generating fake videos and delivering misleading information. It is necessary to develop techniques to detect and watermark the videos generated by machine learning models.

## B  Extending a Diffusion-based I2V Model for Longer Range on LVD-2M

In this section, we present additional qualitative results to demonstrate the effectiveness of fine-tuning a diffusion-based image-to-video (I2V) model.

**Setup.** To compare the effect of LVD-2M to previous datasets on long video generation fine-tuning, we fine-tune the same pretrained diffusion-based I2V model separately on WebVid-10M [3] and LVD-2M. Both datasets are used to fine-tune the model for generating 65-frame videos, with the fine-tuning process running for 20k iterations using identical strategies.

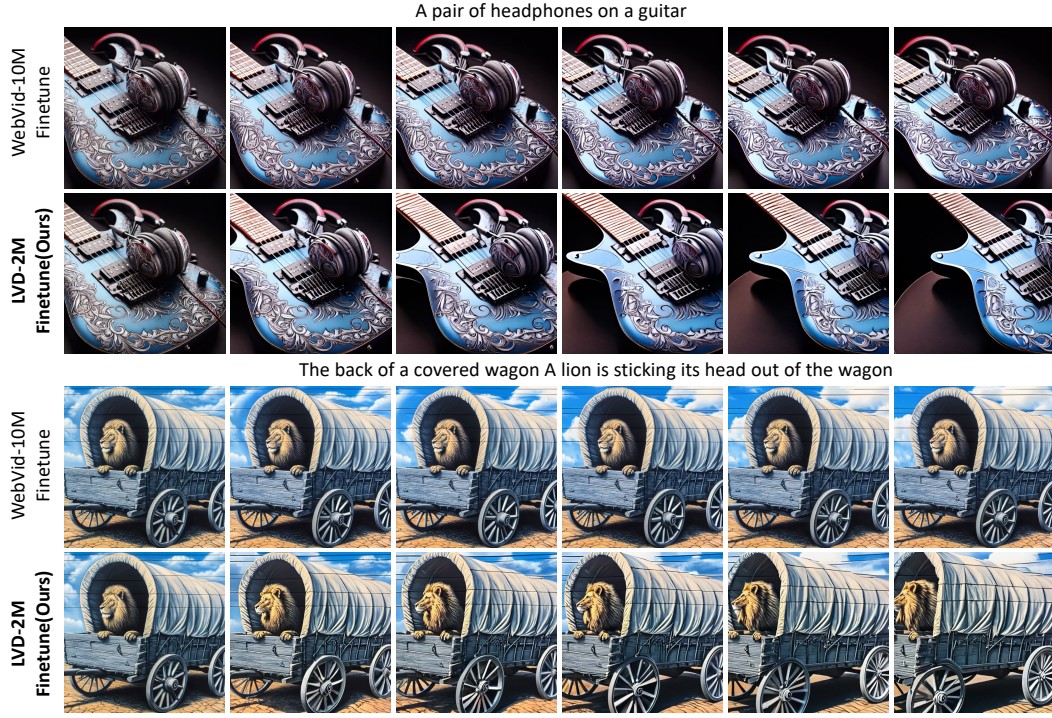

Figure 8: After fine-tuning the diffusion-based I2V model on LVD-2M, the camera perspective will present more translation, compared to WebVid-10M.

**Analysis.** We identify two advantages of fine-tuning with LVD-2M compared to WebVid-10M. First, the camera perspective presents more variation, including translation (Fig. 8) and tracking shots around the main object (Fig. 9). In contrast, after fine-tuning on 65 frames on WebVid-10M, the generated videos are prone to simply repeating the first frame with small variation. Second, there are fewer significant inconsistent transitions after fine-tuning on LVD-2M. As shown in Fig. 10, after fine-tuning on WebVid-10M, the generated videos may abruptly change into white and black mask frames. This phenomenon results from the WebVid training data, where such abrupt transitions are observed for 3D art style videos. For LVD-2M, videos with such transitions are filtered out by our

scene cut detection algorithm. And such cases are less observed in the videos generated by the model fine-tuned on LVD-2M. We also demonstrate I2V results on longer text prompts, as shown in Fig. 11.

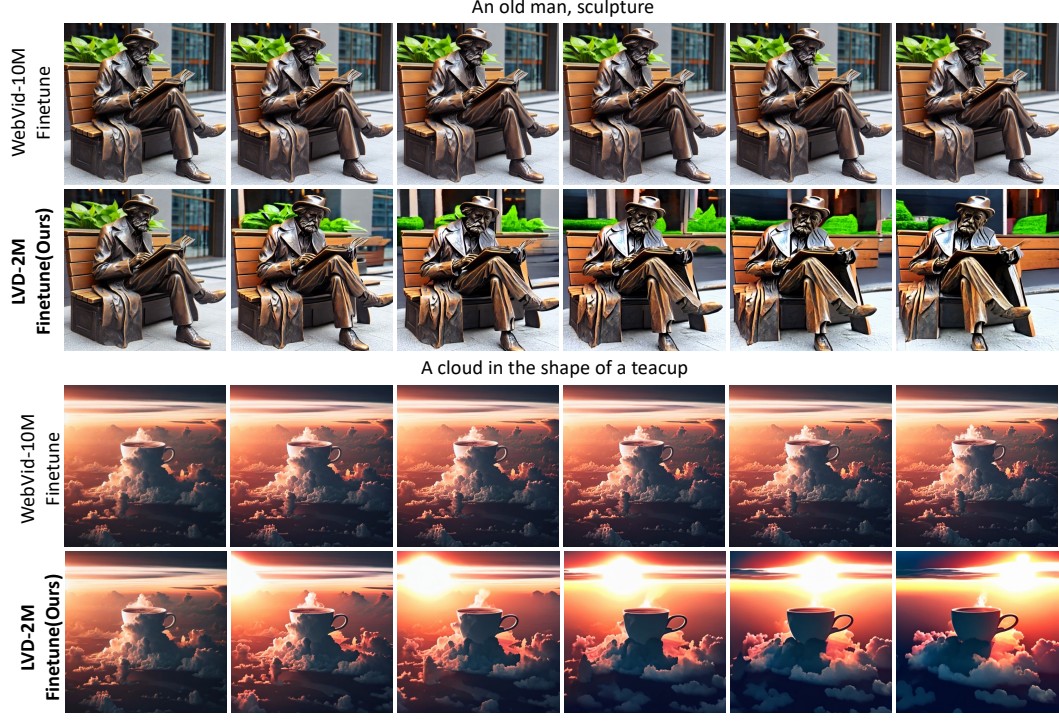

Figure 9: After fine-tuning the diffusion-based I2V model on LVD-2M, the camera view rotates more often and will present more view points, compared to WebVid-10M.

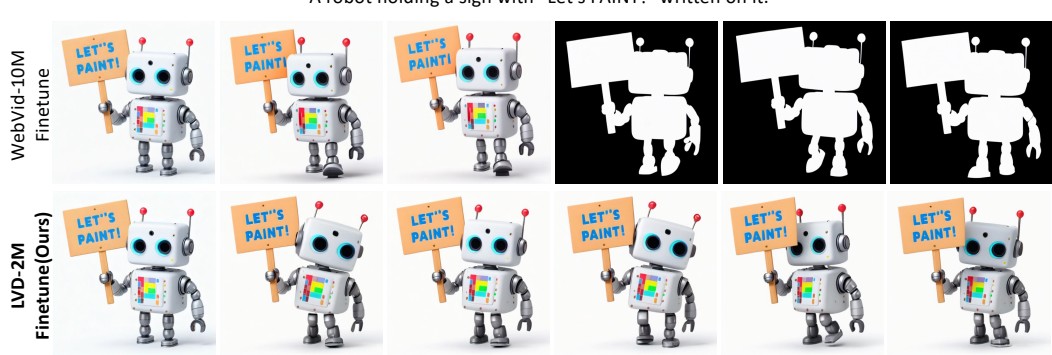

Figure 10: The problem of abrupt transition into black-white mask frames are less observed after fine-tuning the diffusion-based I2V model on LVD-2M.

## C   Qualitative Evaluation for Long Range Video Fine-tuning of LM-based Model on LVD-2M

In this section, we present experiments about generating long videos after fine-tuning the LM-based T2V model on LVD-2M. We choose LM-based model because it can naturally extend the video generation to longer range by directly conditioning on previous generated frames. We also fine-tune the same pretrained LM-based T2V model on WebVid-10M [3] as the baseline.

The video opens with a first-person view from a mountain biker poised at a hill's peak. As he launches downhill, the camera captures the exhilarating rush, the blur of passing trees and rocks. The man, hands gripping the handle bars of the mountain bike, is seen navigating skillfully on the path.

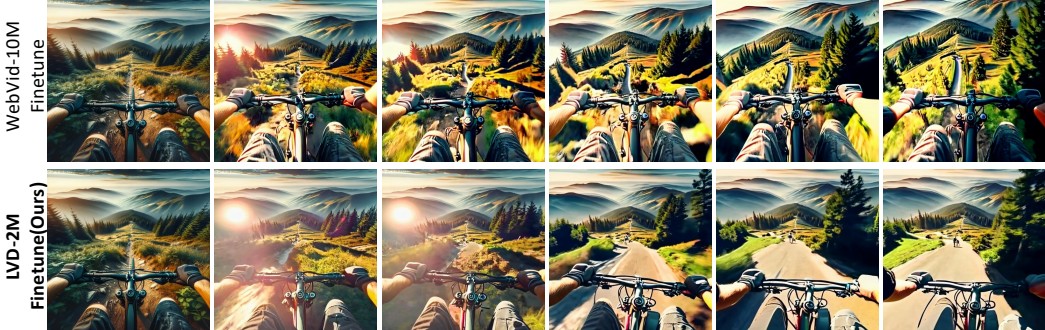

Figure 11: fine-tuning the diffusion-based I2V model on LVD-2M will further improve the capability of the model to generate more dynamic content, compared to WebVid-10M.

**Setup.** We fine-tune the same LM-based model [37] on LVD-2M and WebVid-10M separately on 65 frames (∼10s long) for 10k iterations. Due to a lack of wide accepted long-range video generation benchmark, we choose to qualitatively evaluate the fine-tuned models.

**Analysis.** We provide a comparison of the generated videos from models fine-tuned on LVD-2M and WebVid-10M, as shown in Figure 12. The model fine-tuned on LVD-2M can generate larger motions and more diverse visual elements compared to the one fine-tuned on WebVid-10M. This demonstrates the effectiveness of LVD-2M in enhancing the model's capability to produce highly dynamic and engaging video content.

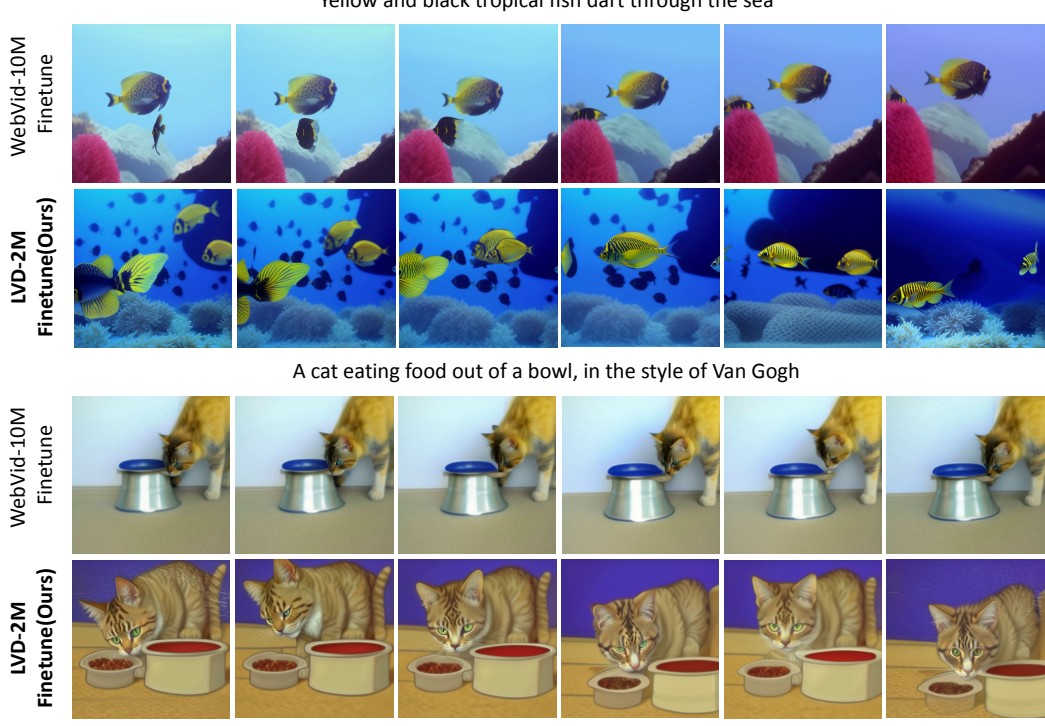

Figure 12: Fintuning the LM-based T2V model on LVD-2M vs. WebVid-10M. After fine-tuning, the model can generate richer content with larger motion. This shows that fine-tuning on LVD-2M can further improve the model's capability to generate more dynamic content, compared to WebVid-10M.

# D   Statistics of LVD-2M and Previous Datasets

In this section, we compare the dataset statistics with the source datasets of ours: WebVid-10M [3], Panda-70M [1], InternVid [22] and HD-VG [2].

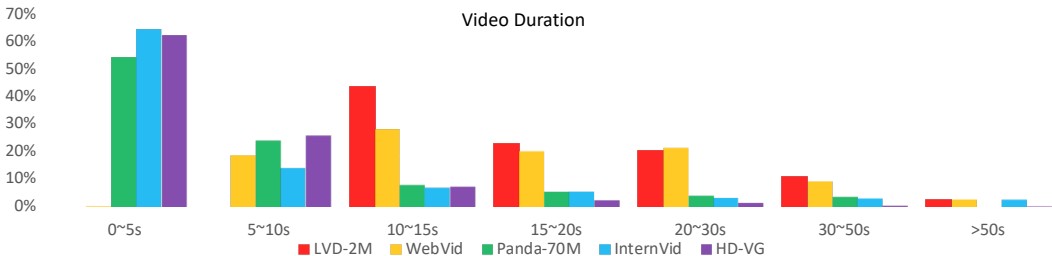

Figure 13: The distribution of video clip duration.

Fig. 13 demonstrates the distribution of duration of the video clips. Among previous datasets, WebVid has larger portion of long videos, mainly because its videos are directly collected from stock footage providers. For other datasets whose videos are from YouTube, short video clips (<10s) almost dominate the datasets. Compared to previous datasets, LVD-2M focuses on video clips longer than 10s, resulting in the collected video clips being significantly longer. This feature of LVD-2M can be useful for learning long-range temporal modeling for video generation.

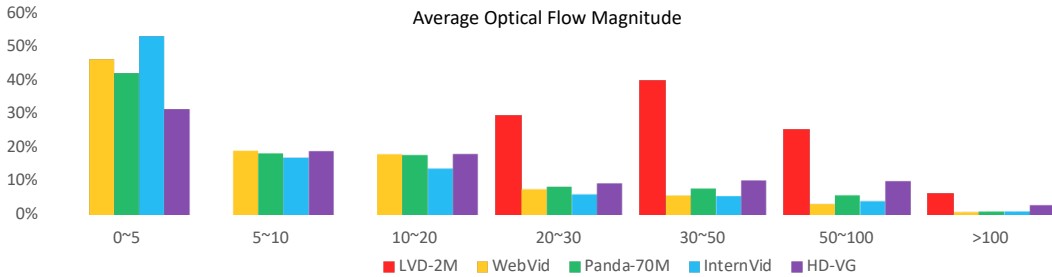

Figure 14: The distribution of average optical flow magnitude. LVD-2M demonstrate significantly larger portion of dynamic (measured by optical flow) videos.

Fig. 14 shows the distribution of optical flow magnitude. Note that this metrics is only calculated for videos longer than 10s. Specifically for calculation, we utilize RAFT [17] with input videos scaled temporally to 2 fps and spatially to $520 \times 960$. The resulting score is the temporal and spatial average of the magnitudes of optical flow estimation. Videos whose average optical flow magnitude is less than 20 are filtered out from our LVD-2M.

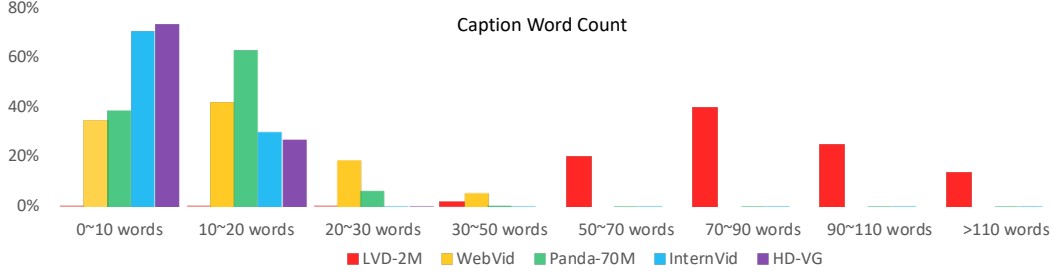

Figure 15: The distribution of caption word count.

Fig. 15 presents the distribution of caption word count. LVD-2M demonstrates a significant gap between previous datasets, with much longer captions. In our captions, we include details about the

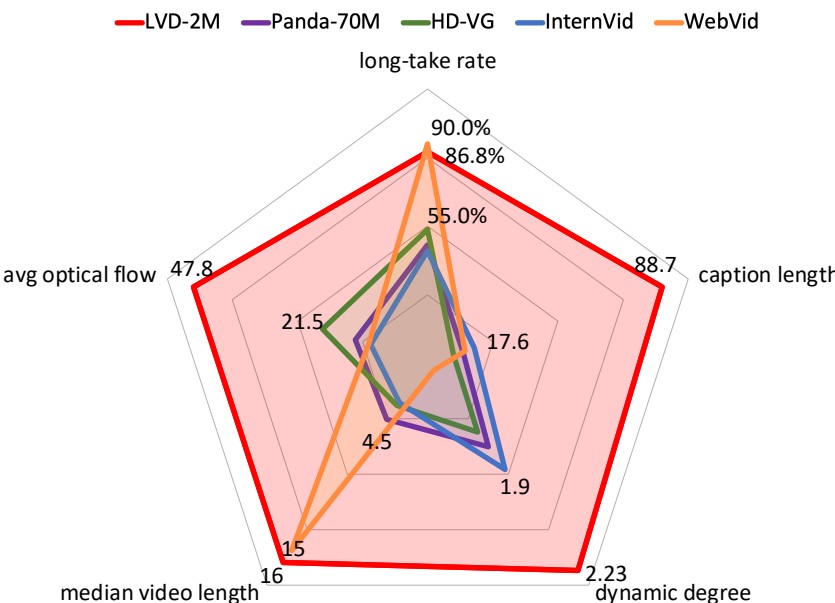

Figure 16: **LVD-2M** presents desirable quality for training of long video generation in 5 dimensions.

actions, characters, camera perspectives and backgrounds. And we employ Claude3-Haiku [21] for refining the captions to be more clear and concise, as we observe much redundancy in the original captions generated by LLaVA-v1.6-34B [20]. As a result, our long captions are both informative and clearly organized.

We further present a radar chart comparing LVD-2M with previous dataset, as shown in Fig. 16. We demonstrate 5 metrics, including the long-take rate measured by human raters, caption length for the average caption word count, dynamic degree which is the average of human rated 1~3 dynamic score, median video clip length and the average optical flow magnitude. For long-take rate, dynamic degree and average optical flow magnitude, the calculation is based on videos longer than 10s. Notably, for the statistics about video clip length, we choose median instead of average here because we find that the average is prone to being affected by a small portion of extremely long video clips. And median video length better reflects the portion of long videos. For the calculation of long-take rate for LVD-2M, in the main paper we exclude the data from WebVid for fair comparison, resulting 77.5%, and here we give the overall long-take rate of LVD-2M, which is 86.8%. LVD-2M presents superior quality compared to previous datasets in various dimensions.

# E   Prompt Designs

## E.1   For PLLaVA: Evaluating Video Quality

The prompts for PLLaVA is presented in Fig. 17. We asks the video LLM to perform binary classification according to different instructions about different aspects of video quality. Only videos considered good according to all defined metrics are kept in our dataset.

## E.2   For LLaVA and Claude3-Haiku: Writing Captions

We present the actual prompts used for our coarse-to-refined caption generation. First, 6 frames sampled from a video clip is concatenated as a 2×3 image grid as the input for LLaVA-v1.6-34B, and the VLM is instructed as in Fig. 18. If there is only one segment from the original video, the generated captions will be refined by Claude3-Haiku [21] as in Fig. 19. When there are multiple consecutive segments from the original video, we use LLaVA-v1.6-34B to generate captions for different segments independently, then we apply Claude3-Haiku for composing the chronologically ordered coarse captions to a refined caption, as shown in Fig. 20.

# F    Discussions on Using MLLM for Data Filtering

In our data pipeline, we utilize PLLaVA [18] for filtering out low-quality video clips, including those with limited content variation or only single-image level semantics. While some previous works [43, 44] meticulously designed methods to distinguish videos or video-question pairs with only single-image level semantics, with recent development of advanced MLLMs [45, 46, 20], we believe evaluation of videos concerning temporal complexity or from other aspects will ultimately be flexibly resolved with proper prompts and powerful MLLMs. However, we also find that current MLLMs are not guaranteed to be capable of video quality evaluation, some of them struggling to follow related instructions. In the future, a comprehensive benchmark for measuring MLLMs capability for video quality evaluation should be helpful to accelerate related research.

Checking content variation.

**USER:**
Evaluate the video using the criterion of content variation:
If the background, setting, and characters are in static states, the video lacks content variation.

If the provided video lacks content variation, you should mark it as "BAD". Otherwise, you should mark it as "GOOD". You must provide a capitalized either "BAD" or "GOOD" answer.

**ASSISTANT(PLLaVA-7B):**
<Answer>

Checking visual diversity and text overlays.

**USER:**
Evaluate the video using these criteria:
1.Visual Diversity: A visually diverse video should have rich content that is visually appealing . If the video is only some person talking to the camera with a static background, it is not diverse. And a video with only texts instead of objects is not diverse.
2.Text Presence: Determine if text overlays dominate the video in a way that detracts from the visual experience.

If the provided video is not visual diverse or having too much text presence, you should mark it as "BAD". Otherwise, you should mark it as "GOOD". You must provide a capitalized either "BAD" or "GOOD" answer.

**ASSISTANT(PLLaVA-7B):**
<Answer>

Figure 17: The prompt used for evaluating video quality with PLLaVA [18].

**USER:**
An image is given containing a 2x3 grid of equally spaced frames sampled from a video. They're arranged in a temporal order from left to right, and then from top to down, all separated by white borders.
Your task is to describe the overall content and context of the video based on the image.
Make sure your description adheres to the guidelines below:
1. Don't describe the content frame-by-frame. Don't use words like 'in the first frame'. Instead, provide an overview of the video that captures details of the main actions, settings, and characters.
2. You should highlight details of any significant events, characters, backgrounds or objects that appear throughout the video.
3. In your description, remember to carefully check the camera perspective, view, movements and changes in shooting angles in the sequence of video frames.

**ASSISTANT(LLaVA-v1.6-34B):**
<Answer>

Figure 18: The prompt used for instructing LLaVA-v1.6-34B [20] to generate relatively coarse captions for video clips.

**USER:**
I need assistance rewriting captions for a video. The new caption should replicate the style typically used in text prompts for video generation.
And your task is to craft a caption that is clear, concise, and factual, following the guidelines below:
1. Describe only what can be directly observed in the video, using straightforward and objective language. In your caption, avoid subjective interpretations or emotional language.
2. Your new caption should provide an overview of the video that captures the main actions, background, visual style, and characters.
3. Organize your caption in a way that effectively and succinctly conveys the storyline or main events of the video.
4. Ensure your caption includes details about the setting, characters and key actions of the video.
5. Don't include any information about the exact number of frames in the video.
6. Do not describe each frame individually. Do not reply with words like 'the first/second/... frame'.

Start your revised caption with the prefix "CAPTION:" and make sure it adheres to the above guidelines.
Here is the raw caption you need to rewrite:
<RAW_CAPTION>

**ASSISTANT(Claude3-Haiku):**
<Answer>

Figure 19: The prompt used for instructing Claude3-Haiku [21] to refine the single coarse caption from LLaVA-v1.6.

**USER:**
I need assistance composing and rewriting captions for a video. The new caption should replicate the style typically used in text prompts for video generation.
And your task is to craft a caption that is clear, concise, and factual, according to given list of cpations.
The list of captions are in a chronological order, describing the content of consecutive video clips from the same video.
When writing the caption, you should follow the guidelines below:
1. Describe only what can be directly observed in the video, using straightforward, concise and objective language. In your caption, avoid subjective interpretations or emotional language.
2. Your new caption should provide an overview of the video that captures the main actions, background, visual style, and characters.
3. Organize your caption in a way that effectively and succinctly conveys the storyline or main events of the video.
4. Ensure your caption concisely includes the details about the setting, characters and key actions of the video.
5. Don't include any information about the exact number of frames or clips in the video.
6. Do not describe each frame individually. Do not reply with words like 'the first/second/... frame'.

Start your revised caption with the prefix "CAPTION:" and make sure it adheres to the above guidelines.
Here is the list of descriptions of video clips:
<DESCRIPTIONS>

**ASSISTANT(Claude3-Haiku):**
<Answer>

Figure 20: The prompt used for instructing Claude3-Haiku [21] to compose the multiple coarse captions from LLaVA-v1.6.

# G   Author Statements

The dataset is open and the data is collected from publicly available resources. For using this dataset, please check for the related license[1]. For the released data records and dataset documentation, please check our homepage at https://github.com/SilentView/LVD-2M.

---

[1] https://raw.githubusercontent.com/microsoft/XPretrain/main/hd-vila-100m/LICENSE

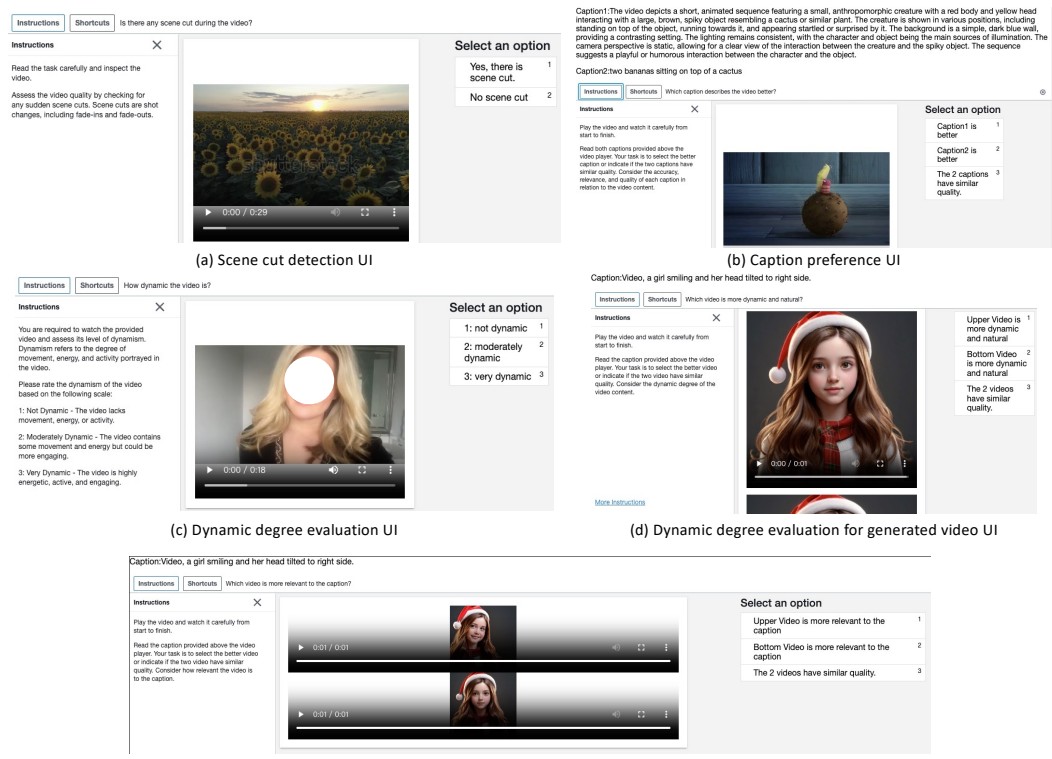

Figure 21: The UI for all the user studies conducted in this work.