# OpenReview forum: "LVD-2M: A Long-take Video Dataset with Temporally Dense Captions"
_NeurIPS.cc/2024/Datasets_and_Benchmarks_Track — NeurIPS 2024 Track Datasets and Benchmarks Poster_

### Official Review · Reviewer_28wF · 2024-07-16
**Good video dataset paper, well-filtered and captioned**

**Rating:** 7
**Confidence:** 5
**Clarity:** The paper is well written.

**Review:**

Some parts in this paper can be better clarified but the overall quality of this paper is good. Originality is not significant since the dataset is a filtered and captioned subset from four existing dataset. However, I believe this dataset has significance meaning to the research community since many video-related tasks such as video generation is bottlenecked by a high quality dataset.

Pros are discussed in other sections.

Cons:
1. Some parts of the paper can be improved for clarity such as:
- It's not clear how the author uses MLLMs to filter out low-quality videos. Specifically, in L154-155 "we filter out videos that lack diversity, lack content variations, or with low perceptual qualities."
- to validate the fairness of user study, a screenshot of the user study would be helpful.
2. This work uses image captioner with an image grid as input, it's unclear why the author chooses such a design when there are existing video captioners[1,2,3]. Intuitively, caption the image grid will generate weak motion description since it's can only obtained by the guesses between frames. The captions are more like describing each frame instead of a storytelling of a consistent video.
3. Experiments are not sufficient to prove the effectiveness of the dataset. It's very unclear for the reviewer to assess how is the model trained on the proposed dataset is better than the previous dataset. The proof in the paper is very vague and quantitative result is missing.
4. Some common-used filters such as OCR and aesthetics score are not applied in this dataset. Is the dataset good enough on text removal and aesthetics?

[1] https://github.com/THUDM/CogVLM2
[2] https://github.com/DAMO-NLP-SG/VideoLLaMA2
[3] https://llava-vl.github.io/blog/2024-04-30-llava-next-video/

**Strengths:**

1. This dataset will be very valuable for various video tasks such as video generation, video understanding, video-language models.
2. The overall quality of the dataset is good.
3. The proposed filtering pipeline is valid.
4. Captioning makes sense and the result is good.

**Additional Feedback:**

For suggestions on rebuttal check "Opportunities For Improvement" section.
Additional questions:
1. This dataset does not apply widely-used low-level filters such as OCR (in addition to text overlay) and aesthetics predictor. Is it intentional? Or why did the author not apply them?
2. WebVid has heavily watermark which will harm the video generation model trains on it. However, I don't see any discussion around it.

**Correctness:**

The dataset is constructed properly. I've addressed my concerns for this dataset in "Review" section.

**Documentation:**

The author provides a homepage for data records and documentation at "https://github.com/SilentView/LVD-2M" but I got 404 opening this page. I encourage the author to update the link during rebuttal.

**Ethics:**

No ethics concerns in this paper.

**Limitations:**

The author addressed the limitations of dataset scale properly as well as potential negative societal impact.

**Opportunities For Improvement:**

Overall I'd recommend to accept this paper, however, the below has to be addressed. I'd be happy to raise the score if the author provides positive feedback in rebuttal.
1. A discussion between this work and recent work ShareGPT4Video.
2. A updated link to access the dataset and its documentation.
3. Clearly discuss how to use MLLM filtering out low-quality videos.
4. Quantitative evaluations of the finetuning experiments are missing. It's very unclear for the reviewer to assess the claimed effectiveness only by the result of human raters. And human raters can be biased.
5. Better discussion on video captioner vs. image captioner.

**Relation To Prior Work:**

Not really. ShareGPT4Video[1] is a recent high quality video dataset work. Although it's a concurrent work of this paper, I'd still suggest the author to discuss the relationship and difference to ShareGPT4Video.

[1] Chen, Lin, et al. "Sharegpt4video: Improving video understanding and generation with better captions." arXiv preprint arXiv:2406.04325 (2024).

**Summary And Contributions:**

This paper introduces a video dataset that collected from four existing datasets, running scene detection to cut the long videos into consistent short clips, then filtered out the low motion clips based on optical flow. Low quality clips are then further filtered out by a 7B VLM. Captions of the clips are generated by an image captioner with a 2x3 image grid of input frames. The paper shows the statistics of the dataset comprehensively and show the effectiveness of this dataset by finetuning moth t2v and i2v models. The contributions are as follow:
1. A well-filtered and captioned long video dataset that is very useful for video generation research.
2. Dataset filtering has both low-level and semantic-level filters that selects high-quality videos with large motions.
3. Proposed a two-stage captioning pipeline that produces high quality caption.

---

> ### Author Rebuttal · Authors · 2024-08-17
>
> # Response to Reviewer 28wF (1/2)
> ### P1: Details for Video filtering with MLLMs and user studies.
>
>  **For filtering videos with PLLaVA-7B**, we uniformly sample 16 frames from each video as the visual input. We run filtering for two rounds to remove unwanted videos. The first round is about perceptual quality (including visual diversity, quality and text presence), and our prompt design is as below:
>  ```
>  USER:
> Evaluate the video using these criteria:
> 1.Visual Diversity: A visually diverse video should have rich content that is visually appealing. If the video is only some person talking to the camera with a static background, it is not diverse. And a video with only texts instead of objects is not diverse.
> 2.Text Presence: Determine if text overlays dominate the video in a way that detracts from the visual experience.
>
> If the provided video is not visually diverse or has too much text presence, you should mark it as “BAD”. Otherwise, you should mark it as “GOOD”. You must provide a capitalized either “BAD” or “GOOD” answer.
>  ```
>  In the second round, we continue to filter videos marked as "GOOD" in the first round, with respect to content variation, using the prompt:
> ```
> USER:
> Evaluate the video using the criterion of content variation:
> If the background, setting, and characters are in static states, the video lacks content variation.
>
> If the provided video lacks content variation, you should mark it as “BAD”. Otherwise, you should mark it as “GOOD”. You must provide a capitalized either “BAD” or “GOOD” answer.
> ```
> And the videos marked as "GOOD" in the second round will be kept in our dataset.
>
> **For user study details,** please check the attached PDF file to this rebuttal, Figure 1.a-1.e for the UI screenshots of our user studies in the paper.
>
> We will add these details above in the final version to help reproduce our work.
>
>
> ### P2: Discussion on video captioner v.s. image captioner.
>
>
> Firstly, we'd like to note that the listed 3 video LLMs were not available when we were implementing our data pipeline in March and April(e.g. VideoLLaMA 2 was released on 2024.06.03). When choosing the captioning models, we believe the performance for this task is highly relevant to [VideoInstruct benchmark](https://paperswithcode.com/dataset/videoinstruct), which contains lots of open-ended questions asking for detailed descriptions of test videos. The top-ranking available VLLMs at the time we did this work were PLLaAV-34B and IG-LLaVA-v1.6-34B. We chose IG-LLaVA-v1.6-34B because its inference time cost is less than PLLaVA-34B, thus providing better scalability for processing large-scale videos. After captioning each video clip with IG-VLM, the final stage of refining and merging captions with LLMs can further strengthen the motion descriptions based on the relationships between different frames and different clips.
>
>
> ### P3: Lack of quantitative evaluation and comparison against previous dataset.
> We have presented user study for video generation results in Figure 6 for quantitative evaluation. And for comparison against previous dataset, we have conduct qualitative comparison evaluating models finetuned on LVD-2M and WebVid-10M separately in Figure 1-5 in the supplementary file.
>
> We additionally provide evaluation results on VBench-I2V, comparing the diffusion-based I2V model finetuned on our LVD-2M and WebVid (as baseline) respectively with the same settings (learning rate, number of iterations, etc.). 256x256 videos with 65 frames spanning ~6s are sampled for evaluation. The following table shows the evaluation results on VBench-I2V.
>
> | FT Dataset     | Dynamic Degree | Aesthetic Quality | Imaging Quality | Background Consistency |
> |-------------|----------------|-------------------|-----------------|------------------------|
> | LVD-2M (ours)      | 30.09%         | 65.00%            | 70.65%          | 94.90%                 |
> | Webvid-10M (baseline) | 21.46%         | 64.57%            | 69.36%          | 96.62%                 |
>
> As shown in the table, models finetuned on LVD-2M present much stronger dynamic degree, better aesthetic quality, and better image quality. However, the "Background Consistency" metric is lower. We identify that this is largely due to the definition of "Background Consistency":
> $$
> S_{background} = \frac{1}{T-1} \sum_{t=2}^{T} \frac{1}{2} (\langle c_{1} \cdot c_{t} \rangle + \langle c_{t-1} \cdot c_{t} \rangle)
> $$
> where $c_i$ represents the CLIP image feature of the $i^{th}$ frame. This metric favors static videos especially for videos with a longer range. Since our dataset features larger dynamics compared to previous datasets, it is reasonable that the model finetuned on LVD-2M presents lower "Background Consistency" but a much higher "Dynamic Degree".

---

> > ### Author Rebuttal · Authors · 2024-08-17
> >
> > # Response to Reviewer 28wF (2/2)
> > ### P4: OCR and aesthetic score for video data filtering.
> >
> > We use PLLaVA-7B to filter out videos with text overlay. We didn't additionally apply OCR after filtering by PLLaVA-7B, because VLLMs could already achieve good accuracy for filtering. VLLMs have achieved high performance recognizing texts from visual contents (e.g. [LLaVA-v1.5](https://llava-vl.github.io/) achieved good results for OCR qualitatively and quantitatively on TextVQA[1]). So the application of VLLMs for filtering videos with text overlay should make other OCR tools unnecessary.
> >
> > For aesthetic scores, we identify that the distribution of video data after being filtered by PLLaVA-7B with our pipeline would be much more preferred by human evaluators, compared to being filtered with aesthetic score estimators (e.g., [LAION-Aesthetics Predictor V2](https://github.com/christophschuhmann/improved-aesthetic-predictor)).
> >
> > To validate the observation, we conduct a user study of the aesthetic level of our dataset and HD-VG dataset. For the user study, we randomly mixed 100 LVD-2M sourced videos (from YouTube) and 100 HD-VG-130M sourced videos and asked users to rate given videos from 1 to 3 considering the aesthetic quality. The screenshots for the user study UI is in Figure 1.f, in the attached PDF file for this rebuttal. The results are as below:
> > |Aesthetic Score by Human     |  1:poor  |  2:fair  |  3:good  |
> > |--------------------------|-----|-----|-----|
> > | HD-VG  |  53% |  42% |   5% |
> > | LVD-2M |  11% |  68% |  21% |
> >
> > As shown in the table, LVD-2M videos have 16% more good videos and 42% less videos with poor aesthetic quality. Because LVD-2M already achieves better human perceptual aesthetic scores than HD-VG which has applied LAION-Aesthetics Predictor V2 for filtering, it should be inefficient to additionally filter videos with existing aesthetic preditors.
> >
> >
> >
> >
> > ### P5: Discussion on the concurrent work ShareGPT4Video
> > For our LVD-2M, the biggest difference from ShareGPT4Video is in the video data collection strategies. LVD-2M features long-take videos with high dynamic degree and perceptual quality, and we meticulously designed algorithms to ensure these features. In contrast, ShareGPT4Video ensures its video quality by algorithms mitigating content homogeneity and by selecting the sources of the videos. For video data collection, our work has more designs to enhance the video quality especially for training video generation models. We will discuss ShareGPT4Video as our concurrent work in the camera-ready version.
> >
> >
> >
> > ### P6: Dataset access.
> > Sorry for the inconvenience. That was because of an unintentional bug. The provided link to our dataset is now publicly available.
> >
> >
> >
> > ### P7: Watermark of WebVid videos.
> > When training video generative models with WebVid videos, we would remove the watermark in the video preprocessing stage. We provide the removal demonstration in Figure 2.a and 2.b in the attached PDF file to this rebuttal. Removing the watermark would largely mitigate the effect of the watermark, and at inference time, watermarked samples are hardly observed.
> >
> > However, because removing the watermarks of videos and distributing the watermark-free videos would constitute copyright violation, we didn't provide the code for watermark removal or discuss about this problem in the paper.
> >
> > [1] Amanpreet Singh, et al., Towards vqa models that can read. In Proceedings of the
> > IEEE/CVF conference on computer vision and pattern recognition, 2019.

---

> > > ### Comment · Reviewer_28wF · 2024-08-18
> > >
> > > I'd like to thank the authors for the rebuttal. The author addressed all my concerns very well, I'm still not fully convinced by the effectiveness to use PLLaVA to assess visual quality of a video, especially the visual diversity, as it seems like a weak signal for MLLM and this work does not include any comparison for some samples. I encourage the authors to give examples in the next version of the paper and validate the effectiveness. I'm happy to raise the score to 7 and recommend to accept the paper.

---

> > > > ### Author Response · Authors · 2024-08-28
> > > >
> > > > Thank you for your thoughtful review. We will continue to refine the paper for the final version according to your suggestions. More validations and samples for the application of Video LLM on video filtering will be included in the camera-ready version.

---

### Official Review · Reviewer_pwtS · 2024-07-22
**A video dataset with auto captions, containing 2M videos, used for finetuning video generation model.**

**Rating:** 5
**Confidence:** 4
**Clarity:** Yes.

**Review:**

Quality:
The paper presents a high-quality piece of research that addresses a clear gap in the field of video generation models. The authors have developed a sophisticated pipeline for filtering and annotating a large-scale video dataset, which is a complex and challenging task.
Clarity:
The paper is well-structured, with a clear abstract, introduction, methodology, experiments, and conclusion.
Originality:
The creation of the LVD-2M dataset, which, to the best of the knowledge, does not have a direct equivalent in terms of scale and the specific focus on long-take videos with temporally dense captions.

**Strengths:**

1. The introduction of the LVD-2M dataset is a significant contribution to the field of video generation. The dataset's unique focus on long-take videos with temporally dense captions fills a gap in available resources for training advanced video generation models.
 2. The paper includes a thoughtful discussion of the potential negative societal impacts of video generation technology, such as the creation of fake videos and misinformation. This demonstrates the authors' commitment to responsible research and their awareness of the broader implications of their work.
3. The technical soundness of the research is evident in the detailed description of the data curation pipeline and the experiments conducted. The paper provides sufficient information for other researchers to understand, replicate, and build upon the work.

**Additional Feedback:**

No additional feedback.

**Correctness:**

The quality of this video dataset is not sure.  Maybe more comparisons with other video datasets on fine-tuning video models will be helpful.

**Documentation:**

Yes.

**Ethics:**

There are no or only very minor ethics concerns.

**Limitations:**

1. The quality of this video dataset is not sure.  Maybe more comparisons with other video datasets on fine-tuning video models will be helpful.
2. The motivation needs to be further clarified. The necessity of constructing LVD-2M is not convincing.

**Opportunities For Improvement:**

1. Some video understanding tasks should be further explored, e.g., video captioning, VQA, rather than video generation.
2. Figure 7 presents some qualitative results in video generation models, but more experiments on quantitative comparisons should also be considered.
3. How to ensure the quality of captions by the proposed pipeline? Compared with captions from human efforts? And the evaluation metrics are not very convincing.

**Relation To Prior Work:**

Yes.

**Summary And Contributions:**

The paper addresses the need for high-quality datasets to train video generation models, particularly for long videos. Most existing models are trained on short video clips, but there is a growing interest in generating longer videos. The authors highlight the limitations of current datasets, which often contain static videos, scene cuts, or sparse annotations. To overcome these challenges, they collected the LVD-2M dataset and introduced the methodology for captioning the videos.

---

> ### Author Rebuttal · Authors · 2024-08-17
>
> # Response to Reviewer pwtS
>
> ### P1: Exploration about more video understanding tasks.
> Our work is motivated by long-take video generation because of the emerging and exploding need for long-take and dynamic video datasets for long video generation in the past year. To our best knowledge, there have not been previous datasets of long-take dynamic videos with temporally-dense captions for long video generation. Such a dataset with high quality will benefit the research community of video generation.
>
> Constructing a long-take video generation dataset is more challenging than constructing a long video understanding dataset. Besides video caption generation, we also need to filter videos based on their dynamic degree, quality, scene cuts, and other criteria as described in the paper to ensure the high quality of the dataset for fine-tuning video generation models.
>
> In practice, the constructed dataset could be applied for video understanding tasks. We leave the explorations to future work.
>
>
> ### P2: More quantitative comparisons for video generation models.
>
> We have presented user study for video generation results in Figure 6 for quantitative evaluation. We additionally provide evaluation results on VBench-I2V, comparing the diffusion-based I2V model finetuned on our LVD-2M and WebVid (as baseline) respectively with the same settings (learning rate, number of iterations, etc.). 256x256 videos with 65 frames spanning ~6s are sampled for evaluation. The following table shows the evaluation results on VBench-I2V.
>
> | FT Dataset     | Dynamic Degree | Aesthetic Quality | Imaging Quality | Background Consistency |
> |-------------|----------------|-------------------|-----------------|------------------------|
> | LVD-2M (ours)      | 30.09%         | 65.00%            | 70.65%          | 94.90%                 |
> | Webvid-10M (baseline) | 21.46%         | 64.57%            | 69.36%          | 96.62%                 |
>
> As shown in the table, models finetuned on LVD-2M present much stronger dynamic degree, better aesthetic quality, and better image quality. However, the "Background Consistency" metric is lower. We identify that this is largely due to the definition of "Background Consistency":
> $$
> S_{background} = \frac{1}{T-1} \sum_{t=2}^{T} \frac{1}{2} (\langle c_{1} \cdot c_{t} \rangle + \langle c_{t-1} \cdot c_{t} \rangle)
> $$
> where $c_i$ represents the CLIP image feature of the $i^{th}$ frame. This metric favors static videos especially for videos with a longer range. Since our dataset features larger dynamics compared to previous datasets, it is reasonable that the model finetuned on LVD-2M presents lower "Background Consistency" but a much higher "Dynamic Degree".
>
>
>
> ### P3: Validation for caption quality and comparison with human efforts.
>
> Firstly, the choice of our captioning model is largely based on performances on [VideoInstrcut Benchmark](https://paperswithcode.com/sota/video-based-generative-performance). This benchmark contains many open-ended detailed video captioning tasks, thus constituting a good validation for caption quality. And we meticulously designed the way to caption minute-level long-take videos by captioning multiple clips separately and summarizing the clips into one consistent caption by LLM, which could ensure better caption quality for long videos.
>
> For further validation, we have compared the captioning quality of our captioning pipeline with other video datasets with human annotations in Figure 5, the right part, in the paper. In this comparison, we asked human evaluators to decide the preferred captions considering the accuracy, relevance, and quality in relation to the video content. Notably, our captions beated WebVid captions by 44% in terms of winning rate. Because the captions of WebVid are human-written captions, the comparison should validate our caption quality, especially in contrast to human efforts.
>
>
> ### P4: Further clarification for the motivation.
>
> Generating consistent long-take videos with large motion and rich content is a research direction that attracts lots of attention in the past year, especially after the release of Sora. However, the previous high-quality video data for training long video generation models are sparse and usually annotated too shortly in previous large-scale video-text pair datasets.
>
> When training models on video clips of longer spans (e.g. >10s), the excessive existence of scene cuts, static videos and perceptually low-quality videos could largely hamper the training efficiency and model performances. Moreover, the too-short captions failing to capture the spatial or temporal details could only provide poor conditions for learning complex long video generation, thus hampering the training effect.
>
> Therefore, the essential motivation of our work is to address the lack of a large-scale dataset consisting of long-take videos with dynamic content and high perceptual quality, paired with temporally dense captions. The proposed LVD-2M will provide an important source of training data for future explorations on long-range video generation.
>
> We acknowledge that long video understanding is also an important research field that needs exploration. However, our work focus more on curating a high-quality dataset for dynamic long-take video generation. Our dataset can be used to build long video understanding benchmarks in the future.

---

> ### Author Response · Authors · 2024-08-28
> **A Kind Reminder to Reviewer pwtS**
>
> Thank you for your thoughtful feedback on our work. We want to remind you that the discussion period is concluding. To facilitate your review, we provide a concise summary below, outlining our responses to each of your concerns:
>
> - P1：We explain that constructing a dataset for long-take video generation could be more challenging than for video understanding. And we leave the explorations for video understanding tasks for feature research.
> - P2: We further conduct an experiment on VBench which quantitatively shows the superiority of LVD-2M against WebVid-10M.
> - P3: We clarify that the caption quality of LVD-2M can be validated by the performances of the chosen captioning VLLM on VideoInstruct Benchmark and our user study in the main paper.
> - P4: We stress the importance of LVD-2M for training long-video generative models, which constitutes our main focus and motivation.
>
> We are eager to confirm whether our responses have adequately addressed your concerns. We look forward to any additional input you may provide.

---

### Official Review · Reviewer_fsrx · 2024-07-25
**Response to Authors**

**Rating:** 7
**Confidence:** 4
**Correctness:** Yes
**Clarity:** Yes

**Review:**

1. The paper addresses the gap in having high-quality long video-text datasets. It identifies key challenges in achieving this including static scenes, scene cuts, and low-quality captions.

2. The data curation pipeline is well-motivated and makes sense. Specifically, I like the adjustments made to the PyScene detection library to increase the coverage, and removing low-quality data with MLLMs.

3. In addition, the paper provides sufficient analysis to show the quality of the data including caption diversity, human evaluation on the dynamic degree and caption quality.

4. I believe that the dataset will be a very useful resource for future works given its 2M scale.

5. I appreciate that the paper evaluates diffusion and autoregressive video generative models.

**Strengths:**

Mentioned in the review

**Additional Feedback:**

No

**Documentation:**

Yes

**Limitations:**

Mentioned in the improvement opportunities.

**Opportunities For Improvement:**

Overall, I believe that the dataset construction part is quite solid but the modeling part of the paper can benefit from further improvements.

1. Having high-quality video-text data should benefit all kinds of video-text applications such as video captioning, video-text understanding (video LLMs), and generation. I am not sure why the authors motivate the work with just the long-form text-to-video generation.

2. The paper does not do a very good job at showcasing the usefulness of the data in a variety of video applications. Even for the video generation aspect, I felt that the evaluation was not comprehensive and suggested that the benefits of the data are minimal if we use a diffusion-based T2V model. Specifically, Diffusion based I2V gets 16% on video-text alignment using LVD-2M in comparison to 14% for the baseline model. With just 50 evaluation prompts (no information about the source of the prompts), the difference between 16% and 14% is just 1 prompt which is too less.

3. Since the paper claims to enhance long-form video-to-text generation, it becomes imperative to understand its importance in comparison to other video-text datasets. The real comparison should not be the base model vs the finetuned model, however, it should be finetuned versions of the model with different datasets vs the finetuned version of the model with LVD-2M trained for the same number of gradient steps.

**Relation To Prior Work:**

The paper should add more citations for the papers that talk about static scenes and temporally low-quality of the video data/benchmarks to better position their paper.

1. https://arxiv.org/abs/2311.10111

2. https://arxiv.org/abs/2206.01720

3. https://aclanthology.org/2023.acl-long.29.pdf

**Summary And Contributions:**

The paper argues that the high-quality video-text data is essential for capable long video generation. To this end, the authors propose a pipeline to curate high-quality video data comprising (a) videos that are longer than 10 seconds, (b) depict a large range of motions (instead of static scenes), and (c) temporally-dense caption generation using a combination of MLLMs and LLMs. The paper is well-written and identifies the existing gaps in the video data, and provides useful fixes to them.

---

> ### Author Rebuttal · Authors · 2024-08-17
>
> # Response to Reviewer fsrx
>
> We appreciate the feedback and suggestions. Below are our replies.
>
> ### P1: Exploration about more video understanding tasks.
> Our work is motivated by long-take video generation because of the emerging and exploding need for long-take and dynamic video datasets for long video generation in the past year. To our best knowledge, there have not been previous datasets of long-take dynamic videos with temporally-dense captions for long video generation. Such a dataset with high quality will benefit the research community of video generation.
>
> Constructing a long-take video generation dataset is more challenging than constructing a long video understanding dataset. Besides video caption generation, we also need to filter videos based on their dynamic degree, quality, scene cuts, and other criteria as described in the paper to ensure the high quality of the dataset for fine-tuning video generation models.
>
> In practice, the constructed dataset could also be applied for video understanding tasks. We leave the explorations to future work.
>
>
>
> ### P2: Experimental validation for the effectiveness of the dataset for video generation.
> For the issue concerning the video-text matching performances of the I2V diffusion models, we'd like to point out that for I2V tasks, the semantics of the prompts are usually already expressed in the given image. Therefore, in Figure 6 it is reasonable that the performance gain against the baseline model for Video Text Matching in I2V task is not obvious. However, the I2V model shows significant improvements on video dynamic degree, demonstrating the effectiveness of our dataset for I2V tasks.
>
> The effectiveness of the dataset for improving video-text matching is demonstrated in the T2V tasks shown in the right part of Figure 6, where the LM-based T2V model finetuned on LVD-2M beats the baseline by 23% in terms of video-text matching. Additionally, we are fine-tuning a T2V diffusion model on LVD-2M and plan to release the evaluation results during discussion period or for the camera-ready version. This could be more direct evidence for the usefulness of LVD-2M to T2V diffusion models.
>
> We also provide the evaluation on VBench in the reply to the next question below.
>
>
>
> ### P3: Compare with baseline datasets.
> We have conducted qualitatively comparisons in Figure 1-5 in the supplementary file as the reviewer suggested: model finetuned on our LVD-2M v.s. model fine-tuned on WebVid-10M for the same number of iterations with the same setting. In addition to the qualitative comparisons in the supplementary file, we additionally provide evaluation results on VBench-I2V, comparing the diffusion-based I2V model finetuned on our LVD-2M and WebVid (as baseline) respectively with the same settings (learning rate, number of iterations, etc.). 256x256 videos with 65 frames spanning ~6s are sampled for evaluation. The following table shows the evaluation results on VBench-I2V.
>
> | FT Dataset     | Dynamic Degree | Aesthetic Quality | Imaging Quality | Background Consistency |
> |-------------|----------------|-------------------|-----------------|------------------------|
> | LVD-2M (ours)      | 30.09%         | 65.00%            | 70.65%          | 94.90%                 |
> | Webvid-10M (baseline)  | 21.46%         | 64.57%            | 69.36%          | 96.62%                 |
>
> As shown in the table, models finetuned on LVD-2M present much stronger dynamic degree, better aesthetic quality, and better image quality. However, the "Background Consistency" metric is lower. We identify that this is largely due to the definition of "Background Consistency":
> $$
> S_{background} = \frac{1}{T-1} \sum_{t=2}^{T} \frac{1}{2} (\langle c_{1} \cdot c_{t} \rangle + \langle c_{t-1} \cdot c_{t} \rangle)
> $$
> where $c_i$ represents the CLIP image feature of the $i^{th}$ frame. This metric favors static videos especially for videos with a longer range. Since our dataset features larger dynamics compared to previous datasets, it is reasonable that the model finetuned on LVD-2M presents lower "Background Consistency" but a much higher "Dynamic Degree".
>
>
>
> ### P4: Prior work about identifying static or temporally low-quality videos.
> We appreciate the suggestions. In the final version, we will add discussions about the 3 works in the related work section.

---

> > ### Comment · Reviewer_fsrx · 2024-08-18
> > **Reply to the rebuttal**
> >
> > I thank the authors for their rebuttal.
> >
> > - As acknowledged, the authors should mention that the datasets can benefit video understanding tasks too in the revised paper.
> >
> > - The unknown nature of the evaluation prompts and a very limited number (just 50 examples) is still a concern. Maybe the authors can add it as a part of their limitation if it is hard to expand the dataset.
> >
> > - Comparison with the baselines: The dataset had three contributions: (a) dynamic degree, (b) long videos without cuts, (c) caption quality. The results that the authors keep showing indicate good benefits in the dynamic degree but the other two contributions are still not validated strongly in the numbers. This is something that the authors can improve upon for the revised paper.
> >
> > I will keep the score. Overall, I like the proposed data curation technique and the authors can revise their paper according to the rebuttal questions/responses.

---

> > > ### Author Response · Authors · 2024-08-28
> > >
> > > Thank you for your constructive feedback. Regarding the scale of test prompts, we believe additionally utilizing VBench which has a larger evaluation prompt set (hundreds of prompts for T2V and I2V) and uses auto metrics for evaluation instead of expensive human labor, could compensate for the relatively smaller prompt scale in the user studies. And we will continue to polish the paper according to your responses.

---

### Official Review · Reviewer_ouB8 · 2024-07-26
**A good paper**

**Rating:** 7
**Confidence:** 5
**Correctness:** Yes
**Clarity:** Yes

**Review:**

Pros:

1. Long-form video generation models face a shortage of high-quality video-text pairs. The proposed large-scale, high-quality dataset is valuable to the community.

2. Technical routines are clear and intuitive. Each step is clearly elucidated and I believe these steps are reasonable and indispensable.

3. The experiments are extensive, including both qualitative and quantitative results. The quality of both the proposed dataset and models fine-tuned by it are evaluated to make the technical contribution more robust.

Cons:

1. For the methodology, I believe some ablations are needed for the expert models adopted for the pipelines. Since there are many alternatives, the rationale for selecting these models needs some discussion.

2. For the experiments, I am curious about the baseline selections. Why do not select some diffusion-based T2V model for evaluation? Does that mean the proposed method is only available for LM-based T2V models? More explanations are needed.

3. Though the authors conduct some human studies to evaluate the quality of the dataset and models, I think some objective metrics are also required. For example, can the proposed pipeline be applied to some video caption benchmarks with long duration? Or if Clipsim or GPT score can be used to evaluate the caption quality? For the T2V and I2V models, can some video generation benchmarks such as VBench or EvalCrafter be applicable?

Overall I think the paper is good and the proposed dataset is valuable for the community. Though some concerns are addressed, I think they can be solved via some revisions. Therefore, I give my original rating as 7. I hope the authors can address these problems above during the rebuttal period.

**Strengths:**

1. Long-form video generation models face a shortage of high-quality video-text pairs. The proposed large-scale, high-quality dataset is valuable to the community.

2. Technical routines are clear and intuitive. Each step is clearly elucidated and I believe these steps are reasonable and indispensable.

3. The experiments are extensive, including both qualitative and quantitative results. The quality of both the proposed dataset and models fine-tuned by it are evaluated to make the technical contribution more robust.

**Additional Feedback:**

N/A

**Documentation:**

Yes

**Limitations:**

The authors have adequately address the limitations and potential negative societal impact of their work yet it won't influence my positive feedback toward this paper.

**Opportunities For Improvement:**

1. More ablations.

2. Using some diffusion-based T2V models as baselines.

3. More objective evaluation.

**Relation To Prior Work:**

Yes

**Summary And Contributions:**

This paper proposes LVD-2M, a large-scale long-take video dataset with temporally dense captions. The proposed dataset features 1) long-form videos with high image quality and large motions, 2) temporally dense captions with rich information and high video-text alignment, and 3) varieties of video duration, caption length, and video categories. The authors fine-tuned an LM-based T2V model and a diffusion-based I2V model using LVD-2M to verify its effectiveness. Both qualitative and quantitative results show the proposed dataset could lead to a higher generation quality for both T2V and I2V models.

---

> ### Author Rebuttal · Authors · 2024-08-17
>
> # Response to Reviewer ouB8
>
> We appreciate your feedback and suggestions. Below are our replies to the questions.
>
>
> ### P1: The rationale for choosing expert models for the pipeline.
> When choosing the expert models, our two considerations are model performance and running speed. High performance ensures the quality of the constructed dataset with our pipeline, and the high inference efficiency of expert models guarantees the scalability of our pipeline for processing large-scale videos.
>
>
> **Rationale for choosing PLLaVA-7B for filtering videos at the semantic level.** We chose the PLLaVA-7B version because that version already satisfies our performance requirements for semantic-level filtering, and it provides higher inference efficiency compared with larger models. We are running an additional experiment to demonstrate the effectiveness of our chosen model in filtering videos based on our criteria. We will report the ablation results during the discussion period.
>
> **Rationale for choosing IG-LLaVA-v1.6-34B for video captioning.** The performance for this task is highly relevant to [VideoInstruct benchmark](https://paperswithcode.com/dataset/videoinstruct), which contains many open-ended questions asking for detailed descriptions of test videos. The top-ranking available VLLMs at the time we did this work were PLLaAV-34B and IG-LLaVA-v1.6-34B. We chose IG-LLaVA-v1.6-34B because its inference time cost is less than PLLaVA-34B, thus providing better scalability for processing large-scale videos.
>
>
> ### P2: The lack of T2V diffusion models for testing.
> We conducted experiments on diffusion-based I2V generation and autoregressive-based T2V generation models to demonstrate the effectiveness of our dataset across different models (diffusion-based and autoregressive-based) and different task settings (I2V and T2V). We didn't additionally train a diffusion-based T2V model because of resource limits. Currently, we are training a T2V diffusion model for validation. We will report the results during the discussion period or in the camera-ready version.
>
>
> ### P3: More objective metrics.
>
>
> **Objective metrics on video generation experiments.** We additionally provide evaluation results on VBench-I2V, comparing the diffusion-based I2V model finetuned on our LVD-2M and WebVid (as baseline) respectively with the same settings (learning rate, number of iterations, etc.). 256x256 videos with 65 frames spanning ~6s are sampled for evaluation. The following table shows the evaluation results on VBench-I2V.
>
> | FT Dataset     | Dynamic Degree | Aesthetic Quality | Imaging Quality | Background Consistency |
> |-------------|----------------|-------------------|-----------------|------------------------|
> | LVD-2M (ours)     | 30.09%         | 65.00%            | 70.65%          | 94.90%                 |
> | Webvid-10M (baseline) | 21.46%         | 64.57%            | 69.36%          | 96.62%                 |
>
> As shown in the table, models finetuned on LVD-2M present much stronger dynamic degree, better aesthetic quality, and better image quality. However, the "Background Consistency" metric is lower. We identify that this is largely due to the definition of "Background Consistency":
> $$
> S_{background} = \frac{1}{T-1} \sum_{t=2}^{T} \frac{1}{2} (\langle c_{1} \cdot c_{t} \rangle + \langle c_{t-1} \cdot c_{t} \rangle)
> $$
> where $c_i$ represents the CLIP image feature of the $i^{th}$ frame. This metric favors static videos especially for videos with a longer range. Since our dataset features larger dynamics compared to previous datasets, it is reasonable that the model finetuned on LVD-2M presents lower "Background Consistency" but a much higher "Dynamic Degree".
>
> **Objective metrics about captioning quality.** For the objective metrics of video clip captioning, as discussed in **P1**, we trust that [VideoInstruct benchmark](https://paperswithcode.com/dataset/videoinstruct) reflects the models' ability in short video clip captioning. Our video clip captioning model IG-LLaVA-v1.6-34B is selected based on the performance on this benchmark. As for long video captioning, to our best knowledge, there are no publically available benchmarks for evaluating temporally-dense captions for long-take videos in the same setting as ours at the time we submit this work. For example, the test videos of MSVD[1] captioning benchmark are usually less than 10s and the ground truth captions are often very simple sentences. Although recent benchmarks like Video-MME[2] could provide long test videos, the VQA tasks are usually about specific aspect of the videos, and open-ended captioning tasks are not supported. The lack of temporally-dense video captions and the difficulties in objective evaluation metrics are the challenges in the field of temporally-dense long video captioning. That's the main reason why we rely mostly on human evaluations for the quality of video captions. As for other alternatives, the captions of LVD-2M are too long (88.7 words on average) to be precisely evaluated by CLIP which accepts texts with a max length of 77, so clipsim may not be an appropriate metric.
>
> [1] Chen D, Dolan W B. Collecting highly parallel data for paraphrase evaluation.//Proceedings of the 49th annual meeting of the association for computational linguistics: human language technologies. 2011: 190-200.
>
> [2] Fu, et al., Video-MME: The First-Ever Comprehensive Evaluation Benchmark of Multi-modal LLMs in Video Analysis. arXiv preprint arXiv:2405.21075

---

> > ### Comment · Reviewer_ouB8 · 2024-08-22
> > **Additional Feedback**
> >
> > I have read the authors' responses and decided to keep my original rating. Besides, experiments on the diffusion-based T2V generation model are important and would strongly support the authors' claim if the proposed dataset could contribute to its generation quality.

---

> > > ### Author Response · Authors · 2024-08-28
> > >
> > > Thank you for your thoughtful review helpful feedback. We are still in the process of training T2V diffusion models for comparisons between LVD-2M and the baseline dataset WebVid-10M. We will add the results to the final version to provide more validations.

---

> > ### Author Response · Authors · 2024-08-28
> > **Ablation for MLLMs Applied to Filter Videos**
> >
> > To quantitatively measure the capability of video LLMs to discriminate high-quality videos from low-quality ones, we manually collect a dataset of 100 videos. In this dataset, 35 videos are classified as "good" in terms of visual diversity, minimal text presence, and dynamic degrees. Other 65 videos are classified as "bad". We instruct the tested video LLMs to classify the quality of the videos with the same text prompts. With the human label being the ground truth, the classification results are as follows:
> >
> > | Model            | Precision (%) | Recall (%) | F1 Score (%) |
> > |------------------|---------------|------------|--------------|
> > | Video-LLaVA-7B[1]      | 61.82         | 97.14      | 75.56        |
> > | LLaVA-OneVision-7B[2] | 35.00         | 100.00     | 51.85        |
> > | PLLaVA-7B           | 75.61         | 88.57      | 81.58        |
> >
> > Among the tested models, LLaVA-OneVision-7B constantly gives a single-word "good" answer, which indicates that it struggles to follow the given instructions for evaluating video quality. According to the results, not all VLLMs are capable of filtering videos. Our current choice, PLLaVA-7B, presents decent performances. In the future, a MLLM finetuned on such task according to human-labeled data may further enhance the filtering effectiveness.
> >
> > [1] Lin B, Zhu B, Ye Y, et al. Video-llava: Learning united visual representation by alignment before projection[J]. arXiv preprint arXiv:2311.10122, 2023.
> >
> > [2] Li B, Zhang Y, Guo D, et al. LLaVA-OneVision: Easy Visual Task Transfer[J]. arXiv preprint arXiv:2408.03326, 2024.

---

### Decision · Program_Chairs · 2024-09-26

**Decision:**

Accept (Poster)

**Comment:**

The paper addresses the difficulties in having high-quality long video-text datasets. It identifies key challenges in achieving this including static scenes, scene cuts, and low-quality captions. And the data curation pipeline is well-motivated and well-designed, which shoudl be help for the video generation task.
Therefore, I am recommending Acceptance of the paper.